Climate-driven disturbances in the San Juan River sub-basin of the Colorado River

5   Katrina E. Bennett[1], Theodore Bohn[2,3], Kurt Solander[1], Nathan G. McDowell[1,5], Chonggang Xu[1], Enrique Vivoni[3,4] and Richard S. Middleton[1]

[1]Earth and Environmental Sciences, Los Alamos National Lab, Los Alamos, NM, 87545

[2]Julie Ann Wrigley Global Institute of Sustainability, Arizona State University, Tempe, AZ, 85287

10  [3]School of Earth and Space Exploration, Arizona State University, Tempe, AZ, 85287

[4]School of Sustainable Engineering and the Built Environment, Arizona State University, Tempe, AZ, 85287

[5] Current affiliation, Pacific Northwest National Laboratory, Richland WA, 99354

Correspondence to: Katrina E. Bennett, kbennett@lanl.gov, Earth and Environmental Sciences, Los Alamos National Lab, Los Alamos, NM, 87545, 505-664-0698

**Abstract**

Accelerated climate change and associated forest disturbances in the Southwestern USA are anticipated to have substantial impacts on regional water resources. Few studies have quantified the impact of both climate change and land cover disturbances on water balances at the basin scale, and none at the regional scale. In this work, we evaluate the impacts of forest disturbances and climate change for a headwater basin to the Colorado River, the San Juan River watershed, using a robustly-calibrated (Nash Sutcliff 0.80) hydrologic model run with updated formulations that improve estimates of evapotranspiration for semi-arid regions. Our results show that future disturbances will have a substantial impact on streamflow with implications for water resource management. Our findings are in contradiction with conventional thinking that forest disturbances reduce ET and increase streamflow. In this study, annual average regional streamflow under the coupled climate-disturbances scenarios is at least 6–11% lower than those scenarios accounting for climate change alone, and for forested zones of the San Juan River basin streamflow is 15–21% lower. The monthly signals of altered streamflow point to an emergent streamflow pattern related to changes in forests of the disturbed systems. Exacerbated reductions of mean and low flows under disturbance scenarios indicate a high risk of lower water availability for forested headwater systems to the Colorado River basin. These findings also indicate that explicit representation of land cover disturbances is required in modelling efforts that consider the impact of climate change on water resources.

# 1    Introduction

Widespread forest disturbances are projected to increase with climate change (McDowell et al., 2016; Allen et al., 2010; Van Mantgem et al., 2009) and this will have major implications for ecosystem services (Anderegg et al., 2013). These ecosystems services impact the provision of food, water, and

energy and therefore necessitate a more robust understanding of how the landscape will respond to the associated shifts in timing and frequency of streamflow and a firm understanding of the dominant processes driving these changes. However, the impacts of coupled disturbances (e.g. climate induced pest outbreaks, fires, drought) in forest ecosystems on water resources remain understudied, despite its importance for natural resource management and energy production in affected basins around the globe.

For instance, climate-induced changes in forests will feedback to the climate by altering the albedo and reducing the carbon sink, which is anticipated to further transform ecosystems in either positive or negative ways (Dale et al., 2001). This is particularly salient for regions of the United States such as the Colorado River basin (CRB) where forest cover is anticipated to be significantly impacted by a higher incidence of wildfire, drought, and pest infestations (Williams et al., 2013).

To date, predictions of future streamflow in forested river basins under future changes in climate and land cover have exhibited wide disagreement as to the strength and even the sign of change. Water yields and peak streamflows in North American river basins are anticipated to either increase, decrease or show no response to changing forest cover (see Table 2 in Adams et al., 2012; Schnorbus et al., 2010; Guardiola-Claramonte et al., 2011; Somor, 2010; McDowell et al., In press, 2017). Causes of the

reported changes have been related to topography (Schnorbus et al., 2010) and climate variability (Allen et al., 2015). Other reported causes are secondary impacts occurring as a result of forest mortality that enhance processes such as radiation (Royer et al., 2011; Varhola et al., 2010), and changes in albedo (Winkler et al., 2010), evapotranspiration (Zou et al., 2010; Kang et al., 2006), groundwater availability (Bearup et al., 2014; Bearup et al., 2016), and soil moisture states (Dale et al., 2001). Snow has been

reported to play an important role in changes in streamflow (Solander et al., 2017; Bennett et al., 2015) through increased snow accumulation and snow melt (Bewley et al., 2010; Boon, 2007), snow cover duration (Boon, 2009) and reduced interception and canopy sublimation and evapotranspiration in

disturbed forests (Livneh et al., 2015b). Temperature changes were noted to play a role in decreasing streamflow in one of eight catchments examined in Somer et al. (2010). Some research points to the fact that we have a limited understanding on why or how streamflow changes in the future under land cover shifts (Bonan, 2008; Buma and Livneh, 2015; Buma and Livneh, 2017).

Disturbances inducing forest loss are poorly represented or entirely absent from earth system models (ESMs) and ecohydrologic models (Brovkin et al., 2013a; Scheller and Mladenoff, 2007; McDowell et al., 2011). In the Coupled Model Intercomparison Project, Version 5 (CMIP5) several earth system models (ESMs) contain dynamic vegetation-cover models (Collins et al., 2011; Watanabe et al., 2011; Brovkin et al., 2013b). ESMs such as those included in CMIP5 model fractional plant functional types

(PFTs) and include feedbacks to the climate and land surface that driven by atmospheric simulations. However, in general, ESMs contain simplifications of the explicit scenario of evolving landscape ecology and do not include a full suite of disturbances (e.g. pests, drought, wildfire), which are extremely difficult to simulate due to the computational expense associated with the coupling between disturbances and the disparate times scales involved. Therefore, current ESMs are limited in their

assessments of the impacts on water resources as well as positive or negative feedbacks driven by the disturbances and the threat of complete system crash e.g. the loss of large tracts of forests, strongly declining water resources (McDowell et al., In press, 2017).
Climate impacts studies from the Southwestern USA highlight the strong influence of changing temperature and precipitation on forest distributions (Dale et al., 2001), forest mortality (Allen et al.,

2015), and streamflow (McCabe and Wolock, 2007; Nash and Gleick, 1991). Specifically, the strong interaction between forests and water (Anderson et al., 1976) means that forest disturbances will have a large impact on water volume and the timing of streamflow. Streamflow decreases were previously reported for the Upper CRB at Lee's Ferry using the Variable Infiltration Capacity (VIC) hydrologic model (Liang et al., 1994; Liang et al., 1996) forced with future climate change scenarios alone and

reported decreased future projected streamflow (Christensen and Lettenmaier, 2002). A separate study reported that CRB streamflow is highly sensitive to precipitation and temperature shifts, with large reductions in streamflow estimated for small increases in precipitation and temperature (Nash and Gleick, 1991) resulting from enhanced rates of evapotranspiration.

The strong link between forests and water, coupled with the emerging threat to future forest viability and integrity, provides a major impetus to study the combined effects of climate and land cover change on streamflow in the CRB. To this end, we apply a hydrologic model that incorporates projected climate changes as well as forest and land cover changes based on recent research. Unlike many other studies, we have included impacts of regrowth of shrubs into our modelling approach since this is the most likely outcome in the CRB (Rother et al., 2015). We determine the direction and shift in streamflow under different scenarios of climate and land cover changes, as well as identify the physical alterations occurring within watersheds across scales and forest compositions, to reveal the driving mechanisms behind the streamflow changes. Knowledge of the physical mechanisms and dominant triggers for streamflow alterations is critical because it will allow decision makers to make better-informed assessments of near- and long-term water operations in a water-constrained world with a changing climate (Zou et al., 2010).

The manuscript is arranged as follows: section 2 details information on forest disturbances and the impact of climate change on streamflow, section 3 outlines our study site and methodology, and section 4, 5 and 6 provide results, discussion, and conclusion, respectively.

## 2       Climate-driven forest disturbances and climate change impacts on streamflow

As fire, pests, and drought modify forests in response to climate change, a number of important energy and water fluxes become altered (Adams et al., 2012). Land covers with a low canopy profile and a small crown cover, e.g. shrubs or bare ground, partition water and energy in different manners from forests that have a large crown cover. For example, shrubs retain snow pack later into the melt season thus snow is able to reach the ground and accumulate (Boon, 2007, 2009). This is largely due to reduced canopy cover which results in reduced interception and sublimation/transpiration (Pomeroy et al., 1998; Bearup et al., 2014) from this above ground storage reservoir, as well as higher solar radiation and wind speeds through the open areas (Harpold et al., 2014). Snow also melts off more quickly in response to increased shortwave radiation and turbulent heat transfers linked to negative longwave radiative fluxes (Burles and Boon, 2011). The responses have also been linked to decreased albedo due to higher litter and/or darker soils associated with dying trees (Bewley et al., 2010; Winkler et al., 2010). Other effects

of tree mortality on the water balance include changes to soil moisture states, changes in groundwater recharge (Bearup et al., 2016), and potential feedbacks to the atmosphere (Bonan, 2008). Variations in the type and magnitude of forest disturbances can also strengthen or dampen some of these effects (Buma and Livneh, 2015). For example, forest wildfires change soil properties and litter

depths, which in turn alter infiltration and runoff processes. In addition, mountain pine beetle infestations change canopy conditions but do not change soil properties and moisture states; and drought may affect the forest canopy and the ability of soils to infiltrate water as trees die and soils desiccate (Adams et al., 2012). Effects may also be impacted by the type of post-disturbance regrowth and extent of disturbance that occurs, such as increases or a difference in shrubs, grasses, forbs coverage

or differences in the composition of tree species that replace the forests (Buma and Livneh, 2017). For instance, water yield declines have been associated with changes in species compositions of forests in the southern Appalachian Mountains (Caldwell et al., 2016). A recent study indicates that soil water residence time is a key factor in water availability post-disturbance owing to limited (or enhanced) evapotranspiration processes (Buma and Livneh, 2017).

Selection of appropriate timescale is an important aspect to consider in vegetation disturbance studies. For example, research focused on forest disturbance over a single summer season, such as post-fire hydrology in New Mexico  shows that surface runoff and recharge both rise following fire (Atchley et al., 2017). However, these studies do not measure the year-long change in water balance, which is critical for snow-driven systems. Examination of water partitioning in disturbed forests within time

frames of less than five years may not adequately resolve effects such as forest succession on the hydrologic regime of study basins (Brown et al., 2005), which may result in a significant overestimation of the impacts of changing forests on water yields over the long term (Pugh and Gordon, 2013). Indeed, most studies that encompass forest disturbance monitoring of greater than five years point to increased evapotranspiration from understory regrowth and an associated decline or mitigating effect of the forest

cover removal (Brown et al., 2014; Biederman et al., 2014; Guardiola-Claramonte et al., 2011). Other research has shown that vegetation management, such as cutting or thinning of forests in the face of climate change, could be used to ameliorate the impacts of reduced streamflow in the CRB, noting that these effects would only last for a period of approximately ten years (Zou et al., 2010).

Finally, spatial heterogeneity also plays an important role in terms of changing hydroecology in the wake of disturbances. Studies that focus on plot-scale results, where the disturbed forest is the primary cover type, illustrate different responses than locations where impacted trees are only a component of the overall land cover, including grasses, shrubs and non-impacted tree species (Biederman et al., 2014;

Pugh and Gordon, 2013). Recent studies examining climate change and extreme wildfire on runoff erosion found that peak streamflow sediment yield will increase with climate change and fire severity due to the lack of spatial heterogeneity in land cover types (Gould et al., 2016). A study by Penn et al. (2016) compared hillslope to watershed scale responses and found a muted effect at the watershed scale in a headwater basin of the Colorado Rocky Mountains.

**3    Methods**

**3.1    Study Site**

To understand the impact of forest disturbances on streamflow under climate changes at different temporal scales and spatial settings, we implemented the VIC model for the San Juan River basin, a sub-basin to the CRB, to simulate streamflow responses to future changes in temperature, precipitation

and land cover. The San Juan is a major headwater basin of the Colorado River, accounting for 15% of streamflow and 22% of the area of the Upper CRB. Spanning four states—the Four Corners—the San Juan watershed is also critical for thermoelectric and hydropower generation, substantial oil and gas development, and extensive irrigated agriculture. Temperature ranges from –2 to 23 °C in January and July, respectively, while average annual precipitation is ~666 mm. The San Juan basin captures the

diversity present across the CRB. For instance, high elevation (> 4000 m) Colorado mountain ranges and large, snowmelt driven rivers comprise the upper San Juan basin. The lower San Juan basin, located in New Mexico and Arizona, is flat, semi-arid and representative of the lower Colorado, with intermittent streams that drain into the main tributary of the San Juan during the summer when they are charged by summer monsoonal rains. The San Juan river eventually drains into the Colorado, just below

the town of Bluff, Utah (Fig. 1).

### 3.2 Hydrologic Model

For this work, we used the Variable Infiltration Capacity (VIC) model version 4.2 (Bohn and Vivoni, 2016; Liang et al., 1994) at a 1/16$^{th}$ degree (6 km) spatial resolution. In each grid cell, VIC simulates vertical energy and water dynamics at an hourly time step for a mosaic of land cover tiles underlain by a

3-layer soil column. Sub-grid heterogeneity in infiltration is represented by a statistical distribution (the variable infiltration capacity curve). Surface runoff is generated via saturation excess, while sub-surface runoff is characterized by the non-linear baseflow curve of Francini and Pacciani (1991). VIC v4.2 includes fractional canopy coverage derived from normalized difference vegetation index (NDVI) and a spatially varying monthly climatology of leaf area index (LAI), albedo, and canopy fraction (Bohn and

Vivoni, 2016). Historical climate data used to run VIC (daily precipitation, minimum and maximum temperature, and wind speed) were obtained from existing gridded data sets for the United States (Livneh et al., 2015a). These daily fields were disaggregated to hourly intervals within the VIC model via algorithms as described in Bohn et al. (2013), which also estimated hourly short- and longwave radiation and humidity. Land cover fractional areas were taken from the average of years 2001-2012 of

the Moderate Resolution Imaging Spectroradiometer (MODIS) MCD12Q1 Collection 5 Plant Functional Type (PFT) product of Friedl et al. (2010), using the International Geosphere-Biosphere Program (IGBP) classification. Repeating climatological seasonal cycles of vegetation parameters (LAI, canopy fraction, and albedo) were derived from the MODIS collection 5 MOD15A2, MCD43A3, and MOD13A1 products (Myneni et al., 2002; Schaaf et al., 2002; Huete et al., 2002) over the period 2000–

2012, and aggregated spatially over the MCD12Q1 land cover classes within each 1/16$^{th}$ degree grid cell.

Soil physical properties (e.g., bulk density, saturated hydraulic conductivity, quartz content) were derived from global datasets such as the United Nations Food and Agriculture Organization (FAO) Digital Soil Map of the World (FAO, 1998). Vegetation structural parameters, such as stomatal and

canopy resistances, were taken from Ducoudré et al. (1993). Several other parameters were calibrated empirically: *D2* and *D3* (the thicknesses of the 2$^{nd}$ and 3$^{rd}$ soil layers); $b_{infilt}$ (the infiltration capacity curve shape parameter); *Ds, Ws, and Dsmax* (non-linear baseflow parameters), and $\alpha_{snow}$ (the albedo of

newly-fallen snow). The model was calibrated using an automatic calibration tool (Yapo et al., 1998) to correct streamflow biases against United States Geological Survey (USGS) naturalized gauged monthly streamflow (2006–2010) for the San Juan River basin at Bluff, UT (Fig. 1, Table 1). Our calibration achieved a Nash Sutcliff efficiency of 0.76 over the calibration period and a 0.60 over the validation period for the San Juan at Bluff, UT USGS monthly naturalized flow data (Table 1).

### 3.3    Climate and Vegetation Change Scenarios

Our study focused on three vegetation projections and four climate projections for a total of 12 different scenarios (Fig. 2). The four climate projections employed to drive VIC were based on Earth System Model (ESM) simulations from CMIP5 (Taylor et al., 2012) climate data including daily temperature, precipitation, and wind speed, downscaled using the Multivariate Adaptive Constructed Analogue (MACA) approach (Fig. 2a and b, Abatzoglou and Brown, 2012), and again, disaggregated to hourly intervals via the algorithms described in Bohn et al. (2013). We selected the four ESMs from CMIP5 because they implemented dynamic vegetation processes: HadGEM2-ES (Collins et al., 2011; Cox, 2001), MIROC-ESM (SIEB-DGVM, Watanabe et al., 2011; Sato et al., 2007), MPI-ESM-LR (JSBACH, Giorgetta et al., 2013; Reick et al., 2013), and IPSL-CM5B-LR (ORCHIDEE, Krinner et al., 2005). We used the representative concentration pathway (RCP) 8.5 emissions scenario, which stipulates strongly increasing emissions by 2100 (Van Vuuren et al., 2011) and corroborates current emissions on par with RCP 8.5 (Le Quéré et al., 2015).

The three vegetation projections used in this study are: (1) *climate-only*, which assumes static land cover (i.e., vegetation types do not change), (2) *dynamic*, which uses the dynamic vegetation changes present within the four CMIP5 ESMs (Fig 2c), and (3) *disturbed*, which uses disturbance projections based on empirically-based statistical estimates of forest mortality in the US Southwest (Fig 2d, McDowell et al., 2016). Vegetation classes were aggregated to six dominant cover types from 16 classes in the IGBP vegetation classification and from 9–13 classes in the ESMs (Table 2). Vegetation changes observed in the *dynamic* and *disturbed* scenarios were applied to the historical MODIS vegetation to alter forest coverage for future runs using a simple delta-change approach. Namely, for

both the ***dynamic*** and ***disturbed*** scenarios, historical forest cover fractions were reduced and concordantly replaced by shrubs in increments of ~10 years (2006–2010, 2010–2020, and so forth); we then ran the projections in 10-year segments, with each segment having a new (constant) forest fraction and starting with the state from the previous time period. Forest cover was reduced by approximately

90% by 2100 for the ***disturbed*** scenario (McDowell and Allen, 2015; McDowell et al., 2015; McDowell et al., 2016) Fig. 2 c, d). Table 3 contains forest and shrub vegetation fractions for each scenario, and LAI, canopy fraction, and albedo for forest and shrubs used in all scenarios from the average of grid cells with greater than 50% forest cover in the San Juan River basin.

We ran the 12 different scenarios for 1950–2099. We analysed daily, seasonal, and annual streamflow

as well as monthly statistics of temperature, precipitation, evapotranspiration, and snow water equivalent (SWE) to understand changes in the water balance in the San Juan River basin. In addition, we investigated the aridity effect upon water availability under forest disturbances using a one-cell analysis. Some studies have suggested an aridity effect, whereby basins with less than 500 mm annual precipitation will see streamflow decrease and vice-versa (Guardiola-Claramonte et al., 2011; Adams et

al., 2012) although this finding is not supported in all work (Brown et al., 2010; Caldwell et al., 2016). For the single cell, we considered climate change by adding +3°C (warm) and +6°C (hot) to the temperature time series and changing precipitation by 20% or -20%. We then changed vegetation characteristics in the single cell to simulate changes in LAI or fractional vegetation spacing (canopy spacing).

**4      Results**

**4.1    Changing climate and land cover**

Temperature and precipitation changes are consistent with previous modelling efforts for the region (Gangopadhyay et al., 2011). The four ESMs projected consistent increases in the annual average temperature (4.3°C–7.1°C, mean change of 5.7°C,) but variable changes in the annual average

precipitation (both increases and decreases, -6.6%–8.2%, mean change of 1.7%) for the San Juan River basin, by comparing the last 30 years of this century to the last 30 years of the previous century. For

both temperature and precipitation, changes are most dramatic and variable after the 2050s, concurrent with increasing greenhouse gas emissions. These four ESMs represent the range of warm/hot and wet/dry changes for the San Juan River basin by the 2080s in CMIP5 (Taylor et al., 2012; Brekke et al., 2013). Seasonal variations in temperature and precipitation change indicate important regional process

shifts. For instance, summer and winter differences show that the summer is warming slightly more than the winter (5.8°C compared to 5.6°C). Annual average fall and winter precipitation is projected to increase while spring and summer precipitation is projected to decrease slightly (-1%) with a large range in variability across the basin. We note that the signals of change for both temperature and precipitation differ from the results offered by the Bureau of Reclamation data sets at 1/8[th] of a degree

and downscaled using a slightly different technique (Bureau of Reclamation, 2011).

Our study identifies a key challenge in representing land cover: land cover change is represented differently by each of the four ESMs due to the variable representations of land use and the application of different dynamic models within each ESM (Arora, 2002). The **_dynamic_** trajectory of change is variable depending on the ESM considered (Fig. 2c). The MIROC-ESM model projects increasing

forest cover, while the HadGEM2-ES model projects the most amount of change in terms of forest loss. Both IPSL-CM5B-LR and MRI-ESM-LR show only a small amount of change in terms of land cover shifts by the 2080s. The **_disturbed_** scenario reflects regional changes expected in the Southwestern USA (McDowell et al. 2016) and therefore projects a more severe, and likely realistic, scenario in terms of projected forest cover change in the San Juan River basin (Fig. 2d).

**4.2   Changing streamflow and water balances**

Annually, simulated streamflow in the San Juan River basin under the **_climate-only_** scenario exhibits differences of -15% to 45%, while the **_disturbed_** scenario indicates a shift of -21% to 34% for the 2080s as compared with historical streamflow, dependent on the ESM (Fig 3). The **_dynamic_** scenario changes the streamflow by -16% to 50%, but is generally very similar to the **_climate-only_** scenario. An exception

to this is the MIROC-ESM **_dynamic_** scenario, which projects an increase in streamflow during winter and a decrease in summer peak flow in response to increasing forest cover and thus decreasing shrubs represented in this model (Fig. 2c). Due to the lack of a large distinction in the vegetation changes under

the *dynamic* scenarios and resultant similarities between the *climate-only* and *dynamic* scenarios, we focus on the differences between the *climate-only* and the *disturbed* scenarios for the remainder of this paper.

Seasonal streamflow in the *climate-only* scenario versus historical simulated streamflow illustrates a shift in the timing of peak flow and increased winter streamflow in the San Juan River basin (Fig 3). Peak streamflow occurs approximately one month earlier, owing to earlier snow melt due to a temperature increase. This shift means that winter flows are higher and also indicates more mid-winter warming events. Under *disturbed* forest cover conditions, seasonal streamflow shows a different hydrograph that represents a shift in timing of winter streamflow and a change in the magnitude of both low (Dec–Jan) and high (Apr–May) streamflow compared to the *climate-only* scenario (green envelope and line, Fig. 3). The VIC simulations driven by *disturbed* scenarios project a lower late fall and winter streamflow, with a delay in spring melt and subsequent increase in the pulse of peak streamflow during April–May. Recessional streamflow (May–Jul) is also slightly higher in the *disturbed scenario* than the *climate-only* scenario, resulting in greater water availability in summer (Fig. 3).

The mechanisms causing these differences in streamflow response to climate change and forest disturbance are illustrated in Figure 4 for forest dominant (more than 50%) regions. In the *climate-only* scenario, the streamflow response is dominated by the impact of temperature on snow pack. Warmer winter temperatures and reduced March snowfall (Fig. 4a) lead to a reduction in snow water equivalent, snow melt, and sublimation from both the pack and the canopy (Fig. 4b-e). Warmer temperatures, earlier snow melt, and greater April rainfall subsequently lead to increases in soil evaporation (Fig. 4f) and transpiration (Fig. 4g) in the spring. Warmer temperatures also lead to increased canopy evaporation in late summer (Fig. 4h). However, the higher rates of soil evaporation and transpiration in the spring deplete the middle layer soil moisture (Fig. 4i), which diminishes their rates in the summer (Fig. 4f, g). In the fall, higher rainfall and warmer temperatures lead to greater soil evaporation but only minimal replenishment of soil moisture. Both bottom layer soil moisture (Fig. 4j) and total (surface and subsurface) runoff (Fig. 4k) exhibit earlier and smaller peaks in the spring, reflecting the earlier melting of the reduced snow pack, and lower levels in summer due to greater ET in the spring.

Two factors in the ***disturbed*** scenario further impact the snow pack, partially compensating for the effects of climate change. First, replacement of forest with shrubs leads to an increase (relative to the ***climate-only*** scenario) in the on-the-ground snow pack accumulation (Fig. 4a), a prominent feature observed in disturbed forests across North America (Boon, 2007; Zou et al., 2010; Biederman et al.,

2015; Brown et al., 2014; Harpold et al., 2014). Shrubs in our VIC simulations have no canopy thus they have no mechanism to intercept snow. This disturbance-driven increase in on-the-ground snow pack partially compensates for the climate-induced streamflow increase during this time of the year. In addition, the larger on-the-ground snow pack and smaller canopy snow pack are accompanied by proportionally higher and lower rates of sublimation from the ground and canopy snow packs,

respectively, relative to ***climate-only*** (Fig. 4c, d). Second, the increase in snow on the ground caused by the higher shrub coverage leads to higher rates of snow melt (approximately 30%, Fig 4c), releasing water during the late spring and early summer (Apr–June, Fig 4k). The larger snow melt flux leads to substantial increases in transpiration in spring and early summer, not only relative to the ***climate-only*** scenario, but also relative to historical conditions (Fig. 4f). A similar increase in soil evaporation (Fig.

4f) does not occur due to the upper and middle soil layers already being saturated in the spring in the ***climate-only*** case (Fig. 4h). The high shrub transpiration rate in the ***disturbed*** scenario delays the larger snow melt flux in reaching the bottom soil layer (Fig. 4i), leading to a delayed peak in runoff (Fig. 4j, k). Meanwhile, LAI values are similar between shrubs and forests (Fig. 4l), indicating that the water and energy balance differences are due mainly to snow processes shifts between those two land cover types.

Differences in streamflow among ***disturbed*** and ***climate-only*** scenarios results are most notable at fine scales (100 km$^2$ to 3,000 km$^2$, 72 grid cells in the San Juan) where the forest cover dominates (more than 50%) the land cover (Fig. 5). As basin size increases (smaller circles in Fig. 5), and forest cover becomes sparser with respect to other types of land cover (e.g. mixed forest, deciduous, shrubs and grass covers), the differences between the ***climate-only*** scenarios and the ***disturbed*** scenarios begin to

decrease, which is corroborated by several other studies (Zhang et al., 2014; Anderegg et al., 2015). The difference between the two scenarios for the densest forest and smallest portion of the basin is approximately 20%, as observed for the larger circles in Fig. 5. This dynamic occurs even under changing precipitation and temperature projections that would otherwise cause increasing or decreasing

streamflow (observed in Fig. 5 colour ramps that change from red to blue for precipitation and different symbols for different temperature ranges). Even as the ESMs project increasing precipitation and temperature, we see the variability in responses through the cascade of scale and land cover variability. This finding is consistent with other studies that observed that variability (forest cover composition and topography) in the area and size of forest-shrub conversion can buffer responses of streamflow or ET shifts from climate change (Winkler et al., 2014; Harpold et al., 2014; Caldwell et al., 2016).

The dynamic between temperature, precipitation, and vegetation changes is examined in more detail by using a simple single cell analysis where climate and vegetation is altered in a sensitivity framework. Fig. 6 illustrates the single cell changes in terms of runoff from the grid cell to investigate the aridity effect. We can see that forests and shrubs act similarly in hot/warm and dry environments while the differences are more pronounced between forests and shrubs in wetter environments (Fig. 6a). For the vegetation changes, we see that changing forest structure (e.g., fraction of the canopy that may occur as a result of pest outbreaks) results in small shifts for forests but very large shifts for shrubs (Fig. 6b). For example, shrubs with 50% canopy spacing produce more water than the forests for the majority of runoff conditions (below the 3rd quantile, Fig 6b). On the other hand, under high runoff conditions (above the 3rd quantile, Fig 6b) shrubs act similarly to regularly spaced shrubs. These changes are on the same order as changes within the climate, generally speaking. Therefore, we hypothesize that larger changes in hydrology under disturbances are more likely to occur not from the forest disturbance itself, but the secondary effects such as regrowth of shrubs, the type of regrowth and the pattern of that regrowth on the landscape. This is why timescale is such an important consideration in studies of this nature, as regrowth patterns are key to understanding how water is partitioned across a disturbed landscape.

## 5    Discussion

Climate change in the CRB is anticipated to cause large impacts to water resources sustainability (Christensen and Lettenmaier, 2007; Rasmussen et al., 2014; Dawadi and Ahmad, 2012). However, to our knowledge few modelling studies have considered the impacts of climate change coupled with changes in vegetation (Buma and Livneh, 2015; Carroll et al., 2017; Pribulick et al., 2016). In this

study, we incorporated changes from the CMIP5 dynamic vegetation models and from an estimate of forest mortality (McDowell et al. 2016) to consider the impacts of both climate and vegetation changes on the water balance in a headwater system to the CRB, the San Juan River basin. We found that failing to consider climate change coupled with vegetation disturbances could result in a ~10% over-estimation

of the annual water availability for this basin. For a river system such as the Colorado that is already gravely stressed, 10% less water in the system may lead to significant water management challenges. And, considering seasonality in flows, the changes we illustrated in our monthly and seasonal scenarios indicate that less water will runoff during spring with more water arriving at peak melt. This could lead to water shortages and flooding and lead to planning issues for short term water delivery.

In this work, we considered not only the impacts of changes to streamflow but also the reasons why streamflow is changing. As in other studies, we found that at the size of the basin and the land cover variability can obscure the signal of change (Biederman et al., 2015; Penn et al., 2016). When we consider forested regions only, we are able to understand how and why streamflow is projected to change under the disturbed conditions. The main mechanism that is shifting streamflow is the manner in

which shrubs impact the water balance during the cold and warm seasons. Snow pack is retained further into the melt season and when snow starts to melt, it melts more quickly and results in higher peak flows. These peak flows, however, occur during a time that the shrubs are using the water, resulting in large transpiration losses. We found differences in canopy evaporation, soil evaporation, and sublimation, but these differences are quite small overall when comparing the overall volume of water

held back in the snowpack (SWE) and leaving via shrub transpiration. Overall, the differences in the mechanisms and delivery of the water changed, and this results in a 20% reduction in the amount of water that is available for runoff through the year, compared to the historical streamflow conditions for the forested regions of the San Juan River basin.

We also found that climate and disturbance have opposing influences on snow pack. In the early

season, the forest disturbance partially compensates for streamflow impacts caused by warmer temperatures (Fig. 3). We see that the shift towards an earlier snowmelt is compensated in part by the fact that shrubs held on to snow for longer into the snow melt season, releasing snow at a later date.

This effect results in overall less water in the system and peak flows that occur on the same point in the year, but are higher in magnitude which results in lower soil moisture, such that the effect of having a late melting pack may not be beneficial for water resources. However, the snow-on-the-ground could have important consequences for early season energy balances.

The findings presented herein represent a deviation from the more broadly accepted viewpoint that forest disturbances will lead to reduced evapotranspiration and increased streamflow. Previous findings tend to be based on studies carried out over short time periods (first 1–2 year responses, < 5 year studies), pair-basin analyses in watersheds disturbed by clear cuts, and where climate variability may have obscured results (Brown et al., 2005). Studies pointing towards increased streamflow also broadly

found evapotranspiration from the canopy decreased, leading to an increase in runoff. However, work based on observations across scales and encompassing multiple disturbances indicates that the regrowth potential for understory, such as shrubs used in this study, is high and that the regrowth is a major control on water availability and direction of change for evapotranspiration and runoff (Caldwell et al., 2016; Biederman et al., 2014; Brown et al., 2014; Biederman et al., 2015; Pribulick et al., 2016).

Moreover, ecologists project that global forest covers are expected to decline and be replaced with species and understory compositions that are more water intensive. It is paramount therefore to treat regrowth correctly within models to align these two currently disparate principles and account accurately for changing streamflow. This is a fundamental issue because ESMs rely upon the research principles developed at plot-scale and watershed-scale observational studies and modelling work.

We also investigated the aridity effect upon water availability under forest disturbances. By comparing vegetation changes to climate shifts in a single cell, we highlighted the impacts of changing temperature and precipitation versus the impacts of changing forest cover properties. We found that the greatest differences in our results for forests and shrubs under climate change occurred under wet conditions, and vice versa for dry conditions. This may be due to the large component of water that is leaving via

evaporation in arid environments. As well, we find an important control on water availability in the shrub environments was the canopy spacing between the shrubs that led to changes in water balances and water partitioning within the environment. Thus, we believe that not only climate (i.e. aridity) but

also vegetation characteristics (i.e. canopy spacing) may play a fundamental role in the impact of disturbances on water availability.

Our study did not incorporate fine-scale processes such as lateral flow, as these processes are not possible using the VIC modelling approach. Suggestions from studies examining vegetation structure

and patchiness illustrates that interconnected hillslope and riparian vegetation, such as shrublands, can receive water from wet upland catchment areas (Thompson et al., 2011). A recent study using the Parflow-CLM model, which incorporates lateral flow processes, found results that are very similar to our study findings (Pribulick et al., 2016). For example, Pribulick et al. (2016) show non-linear declines in streamflow in response to vegetation and temperature changes, with enhanced responses in

snowmelt-driven transects and where percent vegetation change was largest. Other research suggests that water availability declines, that plants adapt and becomes more efficient at using water (Troch et al., 2009), which is not incorporated in the approaches undertaken in this study and Pribulick et al. (2016). Additionally, VIC model parameters that impact changes in water balances have inherent uncertainty under climate change (Bennett et al., In review, 2017), and some of these parameters, such

as canopy overstory, are represented in a binary fashion, which is not necessarily indicative of the forest and shrublands mixtures observed during forest recovery. This is a clear example of improvements that could be made to the modeling approach that should be investigated in future work.

Our study also points to the challenges associated with incorporating spatially variable estimates of changing vegetation patterns. Although CMIP5 contains information from dynamic vegetation models,

we find a large disparity between the results from CMIP5 (***dynamic***) and the current estimates and cutting-edge research on forest cover mortality (***disturbed***); these methods are a scenario-style approach that does not necessary reflect future changing land cover conditions. The importance of accurately representing the effects that are observed at catchment scale and in offline models, such as VIC, cannot be understated. ESMs require the capability to incorporate these changes in a meaningful way that can

be validated using our current understanding of changes to forests in the Southwestern USA, and globally. This will enable the science community to accurately capture the range of responses and the impacts of changing climate and changing land cover on water resources. ESMs such as the Department

of Energy's ACME model have started down this path with the incorporation and testing of the FATES dynamic vegetation model (Fisher R. et al., 2015).

The differences in future streamflow projections in comparison to historical conditions we observed in this study are notable, and suggest that streamflow will decline by a minimum of ~10% for headwater systems of the Colorado. However, in the San Juan River basin forest cover accounts for only 17% of the area, and only 4% of these cells have greater than 80% forest cover. Impacts may be much larger for adjacent systems, such as the Gunnison. Future work will investigate the impacts of changing forest cover in the entire Colorado system, using sensitivity analysis to understand the cumulative effect of changing climate and vegetation cover in the Colorado.

## 6     Conclusion

The implications of declining streamflow due to forest disturbances in the San Juan River basin are significant for the CRB. Given that the San Juan basin is one of the least forested portions of the CRB and assuming similar rates of forest decline will occur nearby (Van Mantgem et al., 2009), other regions of the CRB are likely to experience streamflow declines of much higher severity due to changes in forest cover. Failure to adequately understand the direction and magnitude of changes as we do herein could have catastrophic consequences for those who rely on this resource. The basin—roughly 11% area of the continental United States—directly supports water supply for more than 30 million people, accounts for approximately 15% of the US's crops and livestock (Cohen et al., 2013), and 53 GW of power generation capacity (Buono and Eckstein, 2014; Cohen et al., 2013). Pressing work includes improvements in spatial representation of changing forest covers across the CRB, further investigations into drivers of change including an understanding of the role of aridity and variability, a stronger link between ecological dynamics and hydrological knowledge that is translated into models, and upward propagation of these findings into global and earth system models.

## 7     Acknowledgements

We acknowledge the support of John Abatzaglou and Katherine Hedgwick for providing us additional models not in the original MACA data set. We acknowledge the World Climate Research Programme's

Working Group on Coupled Modelling, which is responsible for CMIP, and we thank the climate modelling groups for producing and making available their model output. For CMIP, the U.S. Department of Energy's Program for Climate Model Diagnosis and Intercomparison provides coordinating support and led development of software infrastructure in partnership with the Global

5    Organization for Earth System Science Portals. Finally, Bennett, Solander, Middleton, McDowell and Xu acknowledge the Los Alamos National Lab's LDRD program for supporting this work. McDowell further acknowledges support of Pacific Northwest National Laboratories LDRD program. Bohn was supported by Grant 1216037 from the U.S. National Science Foundation (NSF) Science, Engineering and Education for Sustainability (SEES) Post-Doctoral Fellowship program and NSF Grant 1462086,

10    DMUU: Decision Centre for a Desert City III: Transformational Solutions for Urban Water Sustainability Transitions in the Colorado River Basin.

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

|  | Objective Function | San Juan at Archuleta | San Juan at Bluff, UT |
|---|---|---|---|
| Calibration (2006-2010) | Volume Bias | 0.91 | 0.90 |
|  | Nash Sutclife | 0.78 | 0.76 |
|  | Nash Sutclife log | 0.77 | 0.75 |
| Validation (2001-2005) | Volume Bias | 0.93 | 0.93 |
|  | Nash Sutclife | 0.83 | 0.60 |
|  | Nash Sutclife log | 0.77 | 0.43 |
| Entire Period (2001-2010) | Volume Bias | 0.90 | 0.91 |
|  | Nash Sutclife | 0.81 | 0.67 |
|  | Nash Sutclife log | 0.78 | 0.58 |

Table 2. IGBP classification names and codes from MODIS, our new classification, and remapped values (RM). Values for each ESM follows, MIROC-ESM, MPI-ESM-LR, HadGEM2-ES, and IPSL-CM5A-LR. For each ESM, the original code and the remapped (RM) values are given.

| IGBP Classification | Code | Our Class | RM | MIROC-ESM | Code | RM | MPI-ESM-LR | Code | RM | HadGEM2-ES | Code | RM | IPSL-CM5B-LR | Code | RM |
|---|---|---|---|---|---|---|---|---|---|---|---|---|---|---|---|
| Evergreen Needleleaf forest | 1 | Evergreen forests | 1 | Tropical Forests | 1 | 1 | glacier | 1 | nc | broadleaf trees | 1 | 2 | Bare soil | 1 | 6 |
| Evergreen Broadleaf forest | 2 | Evergreen forests | 1 | Temperate/ boreal evergreen | 2/4 | 1 | Tropical broadleaf evergreen | 2 | 1 | needleleaf trees | 2 | 1 | Tropical broad-leaved evergreen | 2 | 1 |
| Deciduous Needleleaf forest | 3 | Deciduous Forests | 2 | Temperate/ boreal deciduous | 3/5 | 2 | Tropical broadleaf deciduous | 3 | 2 | C3/C4 grass | 3/4 | 3 | Tropical broad-leaved raingreen | 3 | 2 |
| Deciduous Broadleaf forest | 4 | Deciduous Forests | 2 | C3/C4 grass | 6/7 | 3 | Extra-tropical evergreen | 4 | 1 | shrubs | 5 | 4 | Temperate/boreal needleleaf evergreen | 4/7 | 1 |
| Mixed forest | 5 | Evergreen forests | 1 | Crop | 8 | 5 | extra-tropical deciduous | 5 | 2 | Urban/inland water/bare/ice | 6-9 | 6 | Temperate broad-leaved evergreen | 5 | 1 |
| Closed Shrublands | 6 | Shrubs | 4 | Pasture | 9 | 3 | raingreen shrubs | 6 | 4 | | | | Boreal needleleaf summergreen | 9 | 2 |
| Open Shrublands | 7 | Shrubs | 4 | Bare ground/ Residual | 10/11 | 6 | decidous shrubs | 6 | 4 | | | | Temperate/boreal broad-leaved summergreen | 6/8 | 2 |
| Woody Savannas (40-60% tree cover) | 8 | Grass | 3 | | | | C3/C4 grass | 8/9 | 3 | | | | C3/C4 grass | 10/11 | 3 |
| Savannas (10-40% tree cover) | 9 | Grass | 3 | | | | C3/C4 pasture | 10/11 | 3 | | | | C3/C4 agriculture | 12/13 | 5 |
| Grasslands | 10 | Grass | 3 | | | | C3/C4 crops | 12/13 | 5 | | | | | | |
| Permanent wetlands (assuming like grass) | 11 | Grass | 3 | | | | | | | | | | | | |
| Croplands (corn) | 12 | Crops | 5 | | | | | | | | | | | | |
| Urban and built-up (assuming sparse grass) | 13 | Urban/ Water/ Bare | 6 | | | | | | | | | | | | |
| Cropland/Natural vegetation mosaic | 14 | Crops | 5 | | | | | | | | | | | | |
| Snow and ice (assuming sparse grass) | 15 | Urban/ Water/ Bare | 6 | | | | | | | | | | | | |
| Barren or sparsely vegetated (assuming sparse grass) | 16 | Urban/ Water/ Bare | 6 | | | | | | | | | | | | |

**Table 3. Average values for cells with forest cover greater than 50%. *climate-only* forest and shrub fractional values (top left), *dynamic* forest percentages for each climate model and all decades, and *disturbed* forests and shrubs for each decade. Below, Leaf Area Index (LAI, unitless), albedo (unitless) and canopy spacing (fraction) for all months.**

| climate-only forest | 0.70 | Decade 10 | Decade 20 | Decade 30 | Decade 40 | Decade 50 | Decade 60 | Decade 70 | Decade 80 | Decade 90 | Decade 100 |
|---|---|---|---|---|---|---|---|---|---|---|---|
| climate-only shrubs | (0.08) | | | | | | | | | | |
| *dynamic* forest (HAD) | | 0.69 | 0.69 | 0.68 | 0.67 | 0.65 | 0.63 | 0.61 | 0.59 | 0.57 | 0.56 |
| *dynamic* forest (MIR) | | 0.68 | 0.70 | 0.76 | 0.80 | 0.82 | 0.80 | 0.86 | 0.88 | 0.89 | 0.92 |
| *dynamic* forest (MPI) | | 0.71 | 0.70 | 0.70 | 0.69 | 0.70 | 0.71 | 0.68 | 0.67 | 0.66 | 0.66 |
| *dynamic* forest (IPSL) | | 0.70 | 0.70 | 0.70 | 0.70 | 0.70 | 0.71 | 0.71 | 0.71 | 0.71 | 0.71 |
| *dynamic* forest (HAD) | | 0.09 | 0.10 | 0.10 | 0.12 | 0.14 | 0.16 | 0.18 | 0.19 | 0.21 | 0.23 |
| *dynamic* forest (MIR) | | 0.11 | 0.09 | 0.07 | 0.05 | 0.05 | 0.06 | 0.04 | 0.03 | 0.03 | 0.02 |
| *dynamic* forest (MPI) | | 0.08 | 0.08 | 0.08 | 0.09 | 0.08 | 0.08 | 0.10 | 0.11 | 0.13 | 0.12 |
| *dynamic* forest (IPSL) | | 0.08 | 0.08 | 0.08 | 0.08 | 0.08 | 0.08 | 0.08 | 0.08 | 0.08 | 0.08 |
| disturbed forest | | 0.67 | 0.64 | 0.47 | 0.39 | 0.26 | 0.19 | 0.14 | 0.10 | 0.07 | 0.05 |
| disturbed shrubs | | 0.11 | 0.15 | 0.32 | 0.39 | 0.52 | 0.60 | 0.64 | 0.68 | 0.71 | 0.73 |

HAD=HadGEM2-ES, MIR=MIROC-ESM, MPI=MPI-ESM-LR, IPSL=IPSL-CM5-LR

| Months | Jan | Feb | Mar | Apr | May | Jun | Jul | Aug | Sep | Oct | Nov | Dec |
|---|---|---|---|---|---|---|---|---|---|---|---|---|
| LAI (unitless) | 0.55 (0.43) | 0.54 (0.42) | 0.57 (0.45) | 0.76 (0.66) | 1.20 (1.09) | 2.15 (2.01) | 2.17 (2.06) | 2.07 (1.98) | 1.92 (1.78) | 1.12 (1.04) | 0.75 (0.67) | 0.59 (0.48) |
| Albedo (unitless) | 0.21 (0.28) | 0.22 (0.28) | 0.20 (0.24) | 0.15 (0.17) | 0.11 (0.12) | 0.10 (0.11) | 0.11 (0.12) | 0.11 (0.11) | 0.10 (0.11) | 0.10 (0.11) | 0.12 (0.14) | 0.12 (0.14) |
| Canopy Spacing (fraction) | 0.10 (0.05) | 0.08 (0.04) | 0.09 (0.05) | 0.17 (0.13) | 0.39 (0.33) | 0.68 (0.61) | 0.75 (0.69) | 0.76 (0.70) | 0.69 (0.63) | 0.47 (0.41) | 0.33 (0.25) | 0.15 (0.09) |

1 **9** **Figures**

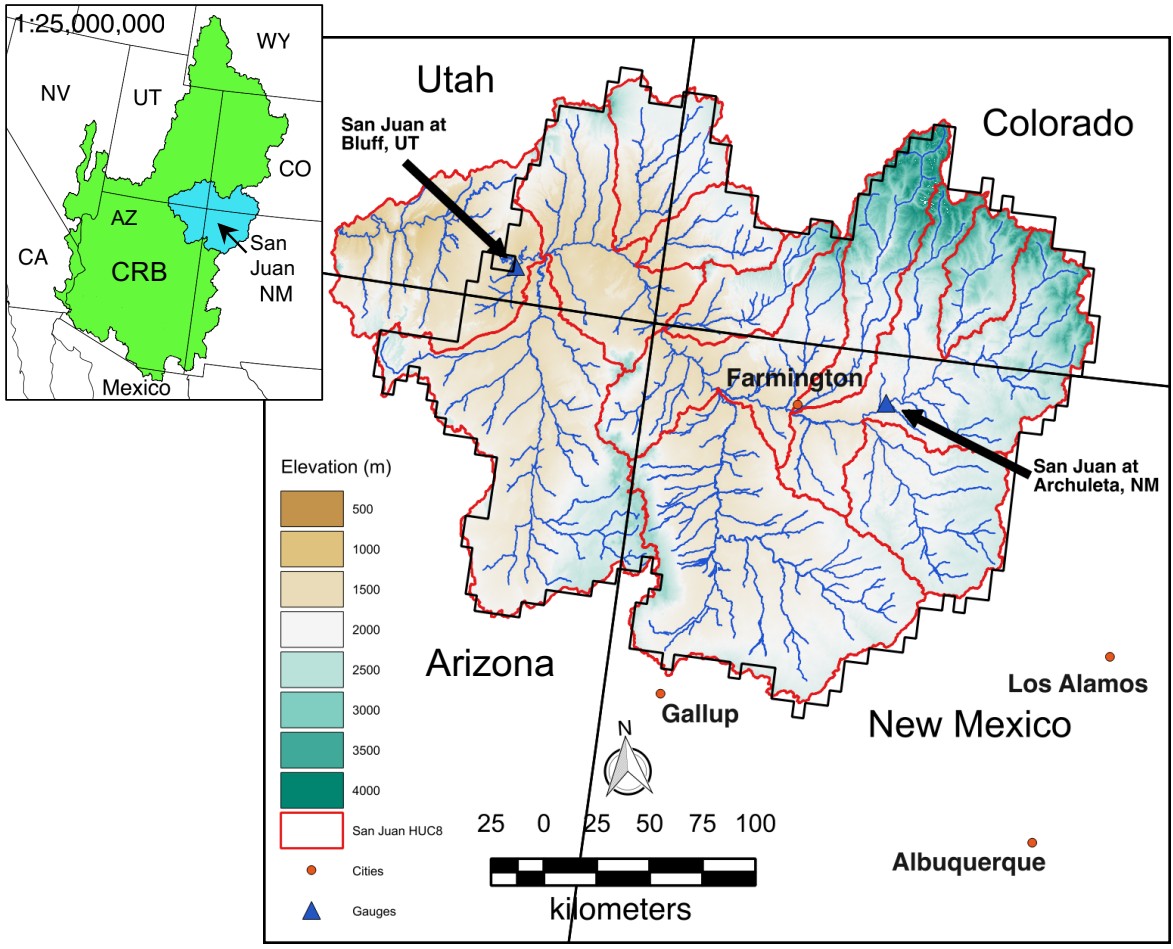

Figure 1. San Juan River basin (61,560 km$^2$) in the Colorado River watershed (634,150 km$^2$, inset), one of the most
important water resources in the western US. The San Juan River basin (outline of HUC8 watersheds shown in red)
spans the Four Corners region (CO, UT, AZ and NM) of the US, and supports multiple energy and water projects.
Naturalized flow gage sites (USBR) at San Juan at Bluff, UT and San Juan at Archuleta, NM are shown with closed blue
triangles.

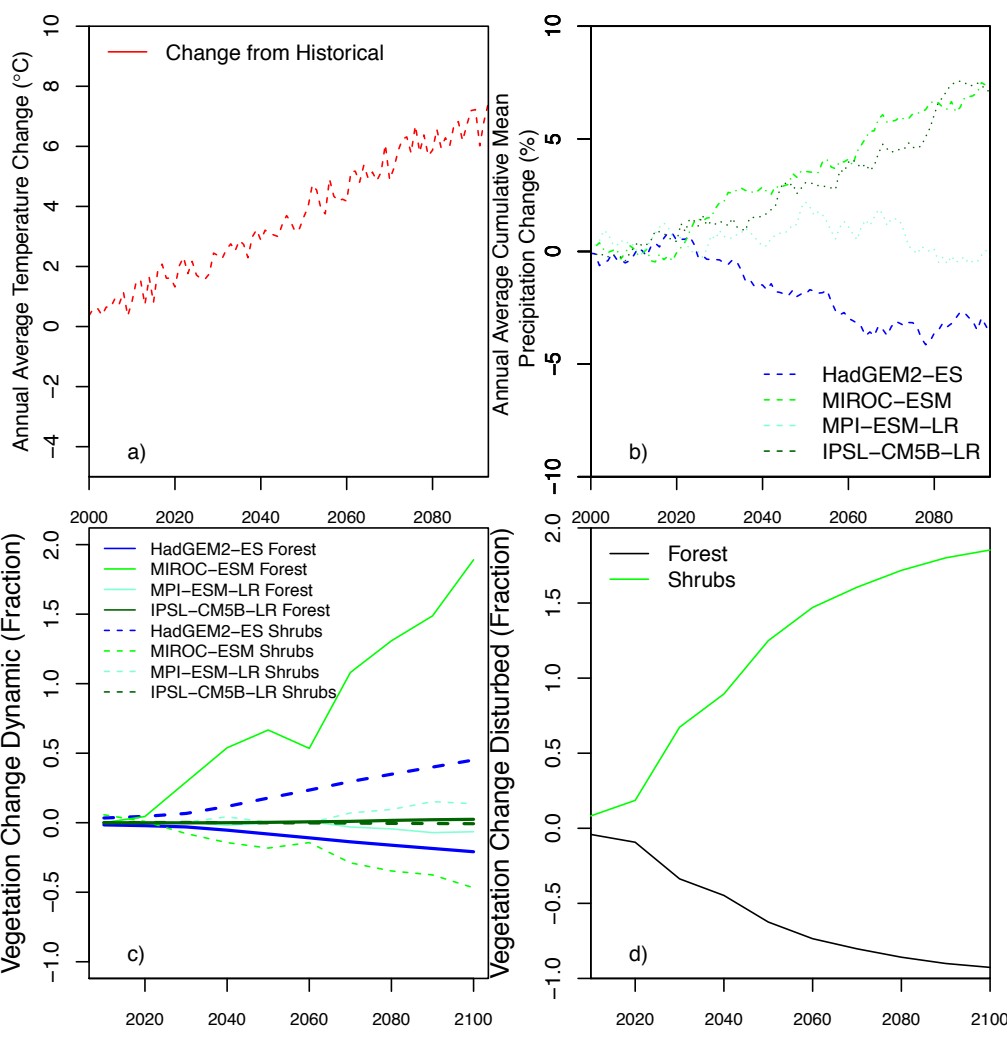

**Figure 2. Temperature (°C, a), precipitation (%, b) and land cover (c and d) changes to the 2080s (2070-2099) averaged over the ESMs and spatially over the San Juan River basin (61,560 km$^2$ or 1570 grid cells). CMIP5 *dynamic* forest disturbances for the four ESMs as a difference from the historical climatology (1970-1999) illustrate more moderate change with HadGEM2-ES projecting the largest decline and MIROC-ESM projecting an increase in forest cover (c). The *disturbed* vegetation scenario based on McDowell et al. 2016 (d) illustrate strongly declining forest covers (>50% forest loss) and increased shrub covers.**

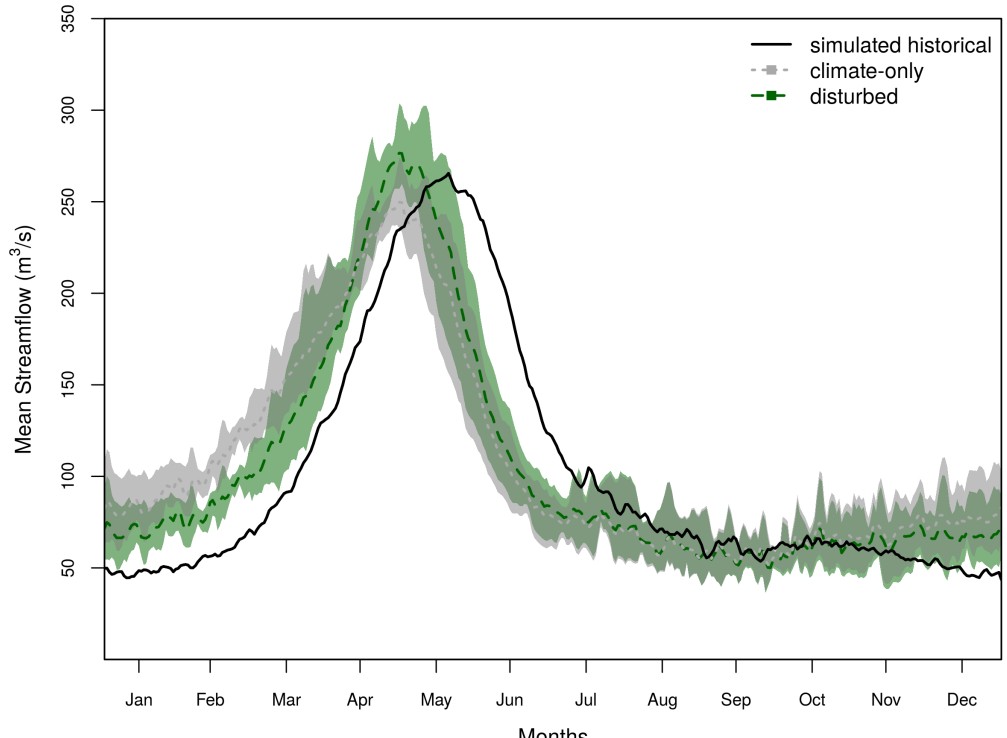

Figure 3. Monthly average future streamflows (m$^3$/s, 2070-2099) compared to historical (1970-1999) for the San Juan
River basin at Bluff, UT. The range of responses for each GCM is represented by the semi-transparent envelope around
the lines. The simulated historical streamflow is shown in black (single line, no envelope), the *climate-only* scenarios are
shown in grey (solid dark grey line, grey envelope), and the *disturbed* scenarios are shown in green (dashed dark green
line, green envelope).

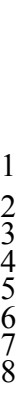

**Figure 4. Monthly water balance panel plot for forested (greater than 50%) regions in the San Juan river basin. Panels**
**include a) precipitation (mm) and temperature (°C) for historical (1970-1999, black) and future (2070-2099, grey), snow**
**water equivalent (SWE, m³/s, b), snow melt (m³/s, c), sublimation from snow (m³/s, d), sublimation from canopy (m³/s, e),**
**soil evaporation (m³/s, f), transpiration (m³/s, g), canopy evaporation (m³/s, h), soil moisture from second soil later (L2,**
**m³/s, i), soil moisture from the third soil layer (L3, m³/s, j), leaf area index (LAI, unitless, k), and runoff (m³/s, l), for**
**simulated historical (black line), *climate-only* (grey line) and *disturbed* scenarios (green line). The months of April and**
**June are indicated with vertical grey lines.**

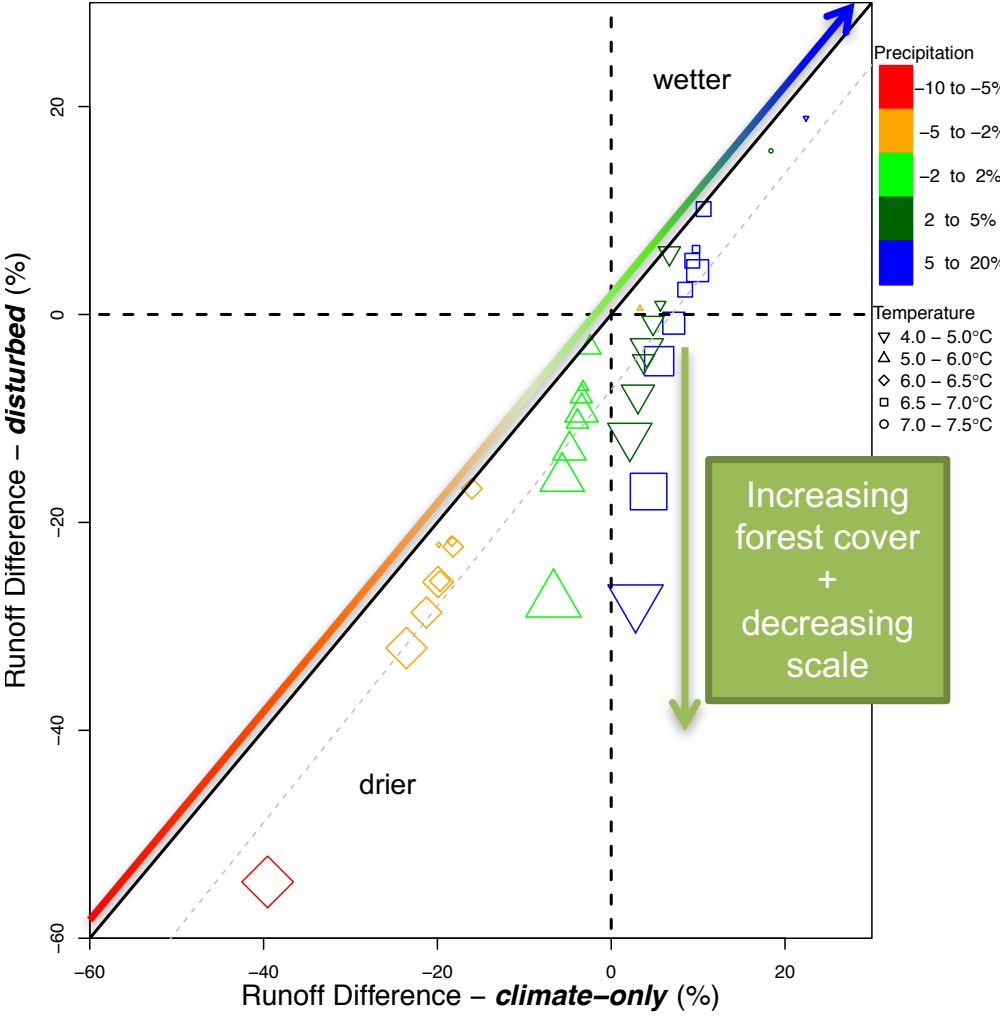

**Figure 5. Runoff differences for the *climate-only* scenario (x-axis) plotted against *disturbed* scenario (y-axis). Increasing sizes indicate results from the entire San Juan River basin (smallest), to all forests (next smallest size), and then forests with greater than 10%, 50%, 70% and 90% coverage (largest). All changes are shown as projected differences from the historical (in % change).**

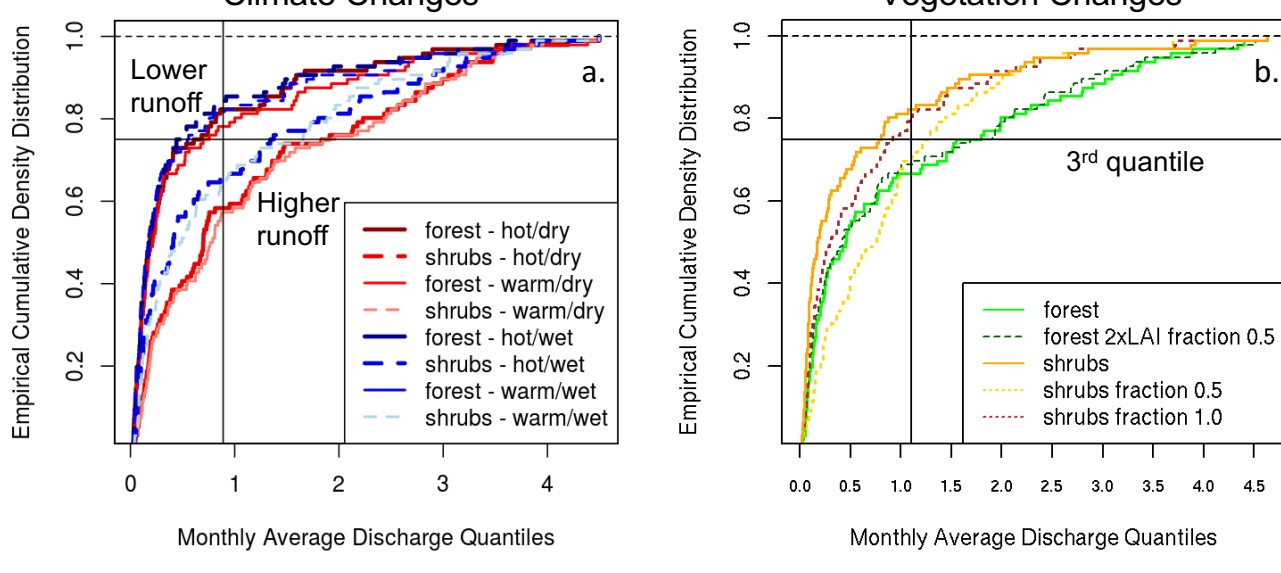

Figure 6. Cumulative density distributions are used here to illustrate the variations in (a) climate and (b) vegetation parameterizations for a single cell in the San Juan River basin. Climate alterations are shown for a hot-wet, hot-dry, warm-wet and warm-dry scenarios for both forest and shrubs. For the vegetation parameterization, forests, forests with two times leaf area index (LAI) and canopy fraction of 0.5, shrubs, and shrubs with fraction of 0.5 and 1.0. A shrub fraction of 1 indicates that there are no spaces between the plants. The third quantile in the average of the distributions is indicated in each plot with intersecting lines.

Climate-driven disturbances in the San Juan River sub-basin of the Colorado River

5  Katrina E. Bennett[1], Theodore Bohn[2,3], Kurt Solander[1], Nathan G. McDowell[1,5], Chonggang Xu[1], Enrique Vivoni[3,4] and Richard S. Middleton[1]

[1]Earth and Environmental Sciences, Los Alamos National Lab, Los Alamos, NM, 87545

[2]Julie Ann Wrigley Global Institute of Sustainability, Arizona State University, Tempe, AZ, 85287

10  [3]School of Earth and Space Exploration, Arizona State University, Tempe, AZ, 85287

[4]School of Sustainable Engineering and the Built Environment, Arizona State University, Tempe, AZ, 85287

[5] Current affiliation, Pacific Northwest National Laboratory, Richland WA, 99354

Correspondence to: Katrina E. Bennett, kbennett@lanl.gov, Earth and Environmental Sciences, Los Alamos National Lab, Los Alamos, NM, 87545, 505-664-0698

## Abstract

Accelerated climate change and associated forest disturbances in the Southwestern USA are anticipated to have substantial impacts on regional water resources. Few studies have quantified the impact of both climate change and land cover disturbances on water balances at the basin scale, and none at the regional scale. In this work, we evaluate the impacts of forest disturbances and climate change for a headwater basin to the Colorado River, the San Juan River watershed, using a robustly-calibrated (Nash Sutcliff 0.80) hydrologic model run with updated formulations that improve estimates of evapotranspiration for semi-arid regions. Our results show that future disturbances will have a substantial impact on streamflow with implications for water resource management. Our findings are in contradiction with conventional thinking that forest disturbances reduce ET and increase streamflow. In this study, annual average regional streamflow under the coupled climate-disturbances scenarios is at least 6–11% lower than those scenarios accounting for climate change alone, and for forested zones of the San Juan River basin streamflow is 15–21% lower. The monthly signals of altered streamflow point to an emergent streamflow pattern related to changes in forests of the disturbed systems. Exacerbated reductions of mean and low flows under disturbance scenarios indicate a high risk of lower water availability for forested headwater systems to the Colorado River basin. These findings also indicate that explicit representation of land cover disturbances is required in modelling efforts that consider the impact of climate change on water resources.

# 1    Introduction

Widespread forest disturbances are projected to increase with climate change (McDowell et al., 2016; Allen et al., 2010; Van Mantgem et al., 2009) and this will have major implications for ecosystem services (Anderegg et al., 2013). These ecosystems services impact the provision of food, water, and

energy and therefore necessitate a more robust understanding of how the landscape will respond to the associated shifts in timing and frequency of streamflow and a firm understanding of the dominant processes driving these changes. However, the impacts of coupled disturbances (e.g. climate induced pest outbreaks, fires, drought) in forest ecosystems on water resources remain understudied, despite its importance for natural resource management and energy production in affected basins around the globe.

For instance, climate-induced changes in forests will feedback to the climate by altering the albedo and reducing the carbon sink, which is anticipated to further transform ecosystems in either positive or negative ways (Dale et al., 2001). This is particularly salient for regions of the United States such as the Colorado River basin (CRB) where forest cover is anticipated to be significantly impacted by a higher incidence of wildfire, drought, and pest infestations (Williams et al., 2013).

To date, predictions of future streamflow in forested river basins under future changes in climate and land cover have exhibited wide disagreement as to the strength and even the sign of change. Water yields and peak streamflows in North American river basins are anticipated to either increase, decrease or show no response to changing forest cover (see Table 2 in Adams et al., 2012; Schnorbus et al., 2010; Guardiola-Claramonte et al., 2011; Somor, 2010; McDowell et al., In press, 2017). Causes of the

reported changes have been related to topography (Schnorbus et al., 2010) and climate variability (Allen et al., 2015). Other reported causes are secondary impacts occurring as a result of forest mortality that enhance processes such as radiation (Royer et al., 2011; Varhola et al., 2010), and changes in albedo (Winkler et al., 2010), evapotranspiration (Zou et al., 2010; Kang et al., 2006), groundwater availability (Bearup et al., 2014; Bearup et al., 2016), and soil moisture states (Dale et al., 2001). Snow has been

reported to play an important role in changes in streamflow (Solander et al., 2017; Bennett et al., 2015) through increased snow accumulation and snow melt (Bewley et al., 2010; Boon, 2007), snow cover duration (Boon, 2009) and reduced interception and canopy sublimation and evapotranspiration in

---

**Deleted:** changes

**Deleted:**

**Deleted:** projected

**Deleted:** or

**Deleted:** changes in

**Deleted:** and

**Deleted:** in disturbed forests

**Deleted:** .

disturbed forests (Livneh et al., 2015b). Temperature changes were noted to play a role in decreasing streamflow in one of eight catchments examined in Somer et al. (2010). Some research points to the fact that we have a limited understanding on why or how streamflow changes in the future under land cover shifts (Bonan, 2008; Buma and Livneh, 2015; Buma and Livneh, 2017).

Disturbances inducing forest loss are poorly represented or entirely absent from earth system models (ESMs) and ecohydrologic models (Brovkin et al., 2013a; Scheller and Mladenoff, 2007; McDowell et al., 2011). In the Coupled Model Intercomparison Project, Version 5 (CMIP5) several earth system models (ESMs) contain dynamic vegetation-cover models (Collins et al., 2011; Watanabe et al., 2011; Brovkin et al., 2013b). ESMs such as those included in CMIP5 model fractional plant functional types

(PFTs) and include feedbacks to the climate and land surface that driven by atmospheric simulations. However, in general, ESMs contain simplifications of the explicit scenario of evolving landscape ecology and do not include a full suite of disturbances (e.g. pests, drought, wildfire), which are extremely difficult to simulate due to the computational expense associated with the coupling between disturbances and the disparate times scales involved. Therefore, current ESMs are limited in their

assessments of the impacts on water resources as well as positive or negative feedbacks driven by the disturbances and the threat of complete system crash e.g. the loss of large tracts of forests, strongly declining water resources (McDowell et al., In press, 2017).
Climate impacts studies from the Southwestern USA highlight the strong influence of changing temperature and precipitation on forest distributions (Dale et al., 2001), forest mortality (Allen et al.,

2015), and streamflow (McCabe and Wolock, 2007; Nash and Gleick, 1991). Specifically, the strong interaction between forests and water (Anderson et al., 1976) means that forest disturbances will have a large impact on water volume and the timing of streamflow. Streamflow decreases were previously reported for the Upper CRB at Lee's Ferry using the Variable Infiltration Capacity (VIC) hydrologic model (Liang et al., 1994; Liang et al., 1996) forced with future climate change scenarios alone and

reported decreased future projected streamflow (Christensen and Lettenmaier, 2002). A separate study reported that CRB streamflow is highly sensitive to precipitation and temperature shifts, with large reductions in streamflow estimated for small increases in precipitation and temperature (Nash and Gleick, 1991) resulting from enhanced rates of evapotranspiration.

The strong link between forests and water, coupled with the emerging threat to future forest viability and integrity, provides a major impetus to study the combined effects of climate and land cover change on streamflow in the CRB. To this end, we apply a hydrologic model that incorporates projected climate changes as well as forest and land cover changes based on recent research. Unlike many other studies,

we have included impacts of regrowth of shrubs into our modelling approach since this is the most likely outcome in the CRB (Rother et al., 2015). We determine the direction and shift in streamflow under different scenarios of climate and land cover changes, as well as identify the physical alterations occurring within watersheds across scales and forest compositions, to reveal the driving mechanisms behind the streamflow changes. Knowledge of the physical mechanisms and dominant triggers for

streamflow alterations is critical because it will allow decision makers to make better-informed assessments of near- and long-term water operations in a water-constrained world with a changing climate (Zou et al., 2010).

The manuscript is arranged as follows: section 2 details information on forest disturbances and the impact of climate change on streamflow, section 3 outlines our study site and methodology, and section

4, 5 and 6 provide results, discussion, and conclusion, respectively.

**2 Climate-driven forest disturbances and climate change impacts on streamflow**

As fire, pests, and drought modify forests in response to climate change, a number of important energy and water fluxes become altered (Adams et al., 2012). Land covers with a low canopy profile and a small crown cover, e.g. shrubs or bare ground, partition water and energy in different manners from

forests that have a large crown cover. For example, shrubs retain snow pack later into the melt season thus snow is able to reach the ground and accumulate (Boon, 2007, 2009). This is largely due to reduced canopy cover which results in reduced interception and sublimation/transpiration (Pomeroy et al., 1998; Bearup et al., 2014) from this above ground storage reservoir, as well as higher solar radiation and wind speeds through the open areas (Harpold et al., 2014). Snow also melts off more quickly in response to

increased shortwave radiation and turbulent heat transfers linked to negative longwave radiative fluxes (Burles and Boon, 2011). The responses have also been linked to decreased albedo due to higher litter and/or darker soils associated with dying trees (Bewley et al., 2010; Winkler et al., 2010). Other effects

| Deleted: 1.1 |
| Deleted: 2 |
| Deleted: data, section 3 details our |

of tree mortality on the water balance include changes to soil moisture states, changes in groundwater recharge (Bearup et al., 2016), and potential feedbacks to the atmosphere (Bonan, 2008). Variations in the type and magnitude of forest disturbances can also strengthen or dampen some of these effects (Buma and Livneh, 2015). For example, forest wildfires change soil properties and litter

depths, which in turn alter infiltration and runoff processes. In addition, mountain pine beetle infestations change canopy conditions but do not change soil properties and moisture states; and drought may affect the forest canopy and the ability of soils to infiltrate water as trees die and soils desiccate (Adams et al., 2012). Effects may also be impacted by the type of post-disturbance regrowth and extent of disturbance that occurs, such as increases or a difference in shrubs, grasses, forbs coverage

or differences in the composition of tree species that replace the forests (Buma and Livneh, 2017). For instance, water yield declines have been associated with changes in species compositions of forests in the southern Appalachian Mountains (Caldwell et al., 2016). A recent study indicates that soil water residence time is a key factor in water availability post-disturbance owing to limited (or enhanced) evapotranspiration processes (Buma and Livneh, 2017).

Selection of appropriate timescale is an important aspect to consider in vegetation disturbance studies. For example, research focused on forest disturbance over a single summer season, such as post-fire hydrology in New Mexico, shows that surface runoff and recharge both rise following fire (Atchley et al., 2017). However, these studies do not measure the year-long change in water balance, which is critical for snow-driven systems. Examination of water partitioning in disturbed forests within time

frames of less than five years may not adequately resolve effects such as forest succession on the hydrologic regime of study basins (Brown et al., 2005), which may result in a significant overestimation of the impacts of changing forests on water yields over the long term (Pugh and Gordon, 2013). Indeed, most studies that encompass forest disturbance monitoring of greater than five years point to increased evapotranspiration from understory regrowth and an associated decline or mitigating effect of the forest

cover removal (Brown et al., 2014; Biederman et al., 2014; Guardiola-Claramonte et al., 2011). Other research has shown that vegetation management, such as cutting or thinning of forests in the face of climate change, could be used to ameliorate the impacts of reduced streamflow in the CRB, noting that these effects would only last for a period of approximately ten years (Zou et al., 2010).

**Deleted:**

**Deleted:** (Atchley et al., 2017),

Finally, spatial heterogeneity also plays an important role in terms of changing hydroecology in the wake of disturbances. Studies that focus on plot-scale results, where the disturbed forest is the primary cover type, illustrate different responses than locations where impacted trees are only a component of the overall land cover, including grasses, shrubs and non-impacted tree species (Biederman et al., 2014; Pugh and Gordon, 2013). Recent studies examining climate change and extreme wildfire on runoff erosion found that peak streamflow sediment yield will increase with climate change and fire severity due to the lack of spatial heterogeneity in land cover types (Gould et al., 2016). A study by Penn et al. (2016) compared hillslope to watershed scale responses and found a muted effect at the watershed scale in a headwater basin of the Colorado Rocky Mountains.

## 3    Methods

### 3.1    Study Site

To understand the impact of forest disturbances on streamflow under climate changes at different temporal scales and spatial settings, we implemented the VIC model for the San Juan River basin, a sub-basin to the CRB, to simulate streamflow responses to future changes in temperature, precipitation and land cover. The San Juan is a major headwater basin of the Colorado River, accounting for 15% of streamflow and 22% of the area of the Upper CRB. Spanning four states—the Four Corners—the San Juan watershed is also critical for thermoelectric and hydropower generation, substantial oil and gas development, and extensive irrigated agriculture. Temperature ranges from –2 to 23 °C in January and July, respectively, while average annual precipitation is ~666 mm. The San Juan basin captures the diversity present across the CRB. For instance, high elevation (> 4000 m) Colorado mountain ranges and large, snowmelt driven rivers comprise the upper San Juan basin. The lower San Juan basin, located in New Mexico and Arizona, is flat, semi-arid and representative of the lower Colorado, with intermittent streams that drain into the main tributary of the San Juan during the summer when they are charged by summer monsoonal rains. The San Juan river eventually drains into the Colorado, just below the town of Bluff, Utah (Fig. 1).

Deleted: For example, recent

Deleted:

Deleted: (Fig. 1)

Deleted: within

Deleted: Upper

Deleted: and

### 3.2 Hydrologic Model

For this work, we used the Variable Infiltration Capacity (VIC) model version 4.2 (Bohn and Vivoni, 2016; Liang et al., 1994) at a 1/16th degree (6 km) spatial resolution. In each grid cell, VIC simulates vertical energy and water dynamics at an hourly time step for a mosaic of land cover tiles underlain by a
3-layer soil column. Sub-grid heterogeneity in infiltration is represented by a statistical distribution (the variable infiltration capacity curve). Surface runoff is generated via saturation excess, while sub-surface runoff is characterized by the non-linear baseflow curve of Francini and Pacciani (1991). VIC v4.2 includes fractional canopy coverage derived from normalized difference vegetation index (NDVI) and a spatially varying monthly climatology of leaf area index (LAI), albedo, and canopy fraction (Bohn and
Vivoni, 2016). Historical climate data used to run VIC (daily precipitation, minimum and maximum temperature, and wind speed) were obtained from existing gridded data sets for the United States (Livneh et al., 2015a). These daily fields were disaggregated to hourly intervals within the VIC model via algorithms as described in Bohn et al. (2013), which also estimated hourly short- and longwave radiation and humidity. Land cover fractional areas were taken from the average of years 2001-2012 of
the Moderate Resolution Imaging Spectroradiometer (MODIS) MCD12Q1 Collection 5 Plant Functional Type (PFT) product of Friedl et al. (2010), using the International Geosphere-Biosphere Program (IGBP) classification. Repeating climatological seasonal cycles of vegetation parameters (LAI, canopy fraction, and albedo) were derived from the MODIS collection 5 MOD15A2, MCD43A3, and MOD13A1 products (Myneni et al., 2002; Schaaf et al., 2002; Huete et al., 2002) over the period 2000–
2012, and aggregated spatially over the MCD12Q1 land cover classes within each 1/16th degree grid cell.

Soil physical properties (e.g., bulk density, saturated hydraulic conductivity, quartz content) were derived from global datasets such as the United Nations Food and Agriculture Organization (FAO) Digital Soil Map of the World (FAO, 1998). Vegetation structural parameters, such as stomatal and
canopy resistances, were taken from Ducoudré et al. (1993). Several other parameters were calibrated empirically: $D2$ and $D3$ (the thicknesses of the 2nd and 3rd soil layers); $b_{infilt}$ (the infiltration capacity curve shape parameter); $Ds, Ws, and Dsmax$ (non-linear baseflow parameters), and $\alpha_{snow}$ (the albedo of

newly-fallen snow). The model was calibrated using an automatic calibration tool (Yapo et al., 1998) to correct streamflow biases against United States Geological Survey (USGS) naturalized gauged monthly streamflow (2006–2010) for the San Juan River basin at Bluff, UT (Fig. 1, Table 1). Our calibration achieved a Nash Sutcliff efficiency of 0.76 over the calibration period and a 0.60 over the validation

period for the San Juan at Bluff, UT USGS monthly naturalized flow data (Table 1).

### 3.3  Climate and Vegetation Change Scenarios

Our study focused on three vegetation projections and four climate projections for a total of 12 different scenarios (Fig. 2). The four climate projections employed to drive VIC were based on Earth System Model (ESM) simulations from CMIP5 (Taylor et al., 2012) climate data including daily

temperature, precipitation, and wind speed, downscaled using the Multivariate Adaptive Constructed Analogue (MACA) approach (Fig. 2a and b, Abatzoglou and Brown, 2012), and again, disaggregated to hourly intervals via the algorithms described in Bohn et al. (2013). We selected the four ESMs from CMIP5 because they implemented dynamic vegetation processes: HadGEM2-ES (Collins et al., 2011; Cox, 2001), MIROC-ESM (SIEB-DGVM, Watanabe et al., 2011; Sato et al., 2007), MPI-ESM-LR

(JSBACH, Giorgetta et al., 2013; Reick et al., 2013), and IPSL-CM5B-LR (ORCHIDEE, Krinner et al., 2005). We used the representative concentration pathway (RCP) 8.5 emissions scenario, which stipulates strongly increasing emissions by 2100 (Van Vuuren et al., 2011) and corroborates current emissions on par with RCP 8.5 (Le Quéré et al., 2015).

The three vegetation projections used in this study are: (1) *climate-only*, which assumes static land

cover (i.e., vegetation types do not change), (2) *dynamic*, which uses the dynamic vegetation changes present within the four CMIP5 ESMs (Fig 2c), and (3) *disturbed*, which uses disturbance projections based on empirically-based statistical estimates of forest mortality in the US Southwest (Fig 2d, McDowell et al., 2016). Vegetation classes were aggregated to six dominant cover types from 16 classes in the IGBP vegetation classification and from 9–13 classes in the ESMs (Table 2). Vegetation

changes observed in the *dynamic* and *disturbed* scenarios were applied to the historical MODIS vegetation to alter forest coverage for future runs using a simple delta-change approach. Namely, for

---

**Deleted:** 90

**Deleted:** 93

**Deleted:** the

**Deleted:** Coupled Model Intercomparison Project, Version 5

both the ***dynamic*** and ***disturbed*** scenarios, historical forest cover fractions were reduced and concordantly replaced by shrubs in increments of ~10 years (2006–2010, 2010–2020, and so forth); we then ran the projections in 10-year segments, with each segment having a new (constant) forest fraction and starting with the state from the previous time period. Forest cover was reduced by approximately

90% by 2100 for the ***disturbed*** scenario (McDowell and Allen, 2015; McDowell et al., 2015; McDowell et al., 2016) Fig. 2 c, d). Table 3 contains forest and shrub vegetation fractions for each scenario, and LAI, canopy fraction, and albedo for forest and shrubs used in all scenarios from the average of grid cells with greater than 50% forest cover in the San Juan River basin.

We ran the 12 different scenarios for 1950–2099. We analysed daily, seasonal, and annual streamflow

as well as monthly statistics of temperature, precipitation, evapotranspiration, and snow water equivalent (SWE) to understand changes in the water balance in the San Juan River basin. In addition, we investigated the aridity effect upon water availability under forest disturbances using a one-cell analysis. Some studies have suggested an aridity effect, whereby basins with less than 500 mm annual precipitation will see streamflow decrease and vice-versa (Guardiola-Claramonte et al., 2011; Adams et

al., 2012) although this finding is not supported in all work (Brown et al., 2010; Caldwell et al., 2016). For the single cell, we considered climate change by adding +3°C (warm) and +6°C (hot) to the temperature time series and changing precipitation by 20% or -20%. We then changed vegetation characteristics in the single cell to simulate changes in LAI or fractional vegetation spacing (canopy spacing).

**4      Results**

**4.1    Changing climate and land cover**

Temperature and precipitation changes are consistent with previous modelling efforts for the region (Gangopadhyay et al., 2011). The four ESMs projected consistent increases in the annual average temperature (4.3°C–7.1°C, mean change of 5.7°C,) but variable changes in the annual average

precipitation (both increases and decreases, -6.6%–8.2%, mean change of 1.7%) for the San Juan River basin, by comparing the last 30 years of this century to the last 30 years of the previous century. For

both temperature and precipitation, changes are most dramatic and variable after the 2050s, concurrent with increasing greenhouse gas emissions. These four ESMs represent the range of warm/hot and wet/dry changes for the San Juan River basin by the 2080s in CMIP5 (Taylor et al., 2012; Brekke et al., 2013). Seasonal variations in temperature and precipitation change indicate important regional process

shifts. For instance, summer and winter differences show that the summer is warming slightly more than the winter (5.8°C compared to 5.6°C). Annual average fall and winter precipitation is projected to increase while spring and summer precipitation is projected to decrease slightly (-1%) with a large range in variability across the basin. We note that the signals of change for both temperature and precipitation differ from the results offered by the Bureau of Reclamation data sets at $1/8^{th}$ of a degree

and downscaled using a slightly different technique (Bureau of Reclamation, 2011).

Our study identifies a key challenge in representing land cover: land cover change is represented differently by each of the four ESMs due to the variable representations of land use and the application of different dynamic models within each ESM (Arora, 2002). The ***dynamic*** trajectory of change is variable depending on the ESM considered (Fig. 2c). The MIROC-ESM model projects increasing

forest cover, while the HadGEM2-ES model projects the most amount of change in terms of forest loss. Both IPSL-CM5B-LR and MRI-ESM-LR show only a small amount of change in terms of land cover shifts by the 2080s. The ***disturbed*** scenario reflects regional changes expected in the Southwestern USA (McDowell et al. 2016) and therefore projects a more severe, and likely realistic, scenario in terms of projected forest cover change in the San Juan River basin (Fig. 2d).

**4.2    Changing streamflow and water balances**

Annually, simulated streamflow in the San Juan River basin under the ***climate-only*** scenario exhibits differences of -15% to 45%, while the ***disturbed*** scenario indicates a shift of -21% to 34% for the 2080s as compared with historical streamflow, dependent on the ESM (Fig 3). The ***dynamic*** scenario changes the streamflow by -16% to 50%, but is generally very similar to the ***climate-only*** scenario. An exception

to this is the MIROC-ESM ***dynamic*** scenario, which projects an increase in streamflow during winter and a decrease in summer peak flow in response to increasing forest cover and thus decreasing shrubs represented in this model (Fig. 2c). Due to the lack of a large distinction in the vegetation changes under

**Deleted:** d

the ***dynamic*** scenarios and resultant similarities between the ***climate-only*** and ***dynamic*** scenarios, we focus on the differences between the ***climate-only*** and the ***disturbed*** scenarios for the remainder of this paper.

Seasonal streamflow in the ***climate-only*** scenario versus historical simulated streamflow illustrates a

shift in the timing of peak flow and increased winter streamflow in the San Juan River basin (Fig 3). Peak streamflow occurs approximately one month earlier, owing to earlier snow melt due to a temperature increase. This shift means that winter flows are higher and also indicates more mid-winter warming events. Under ***disturbed*** forest cover conditions, seasonal streamflow shows a different hydrograph that represents a shift in timing of winter streamflow and a change in the magnitude of both

low (Dec–Jan) and high (Apr–May) streamflow compared to the ***climate-only*** scenario (green envelope and line, Fig. 3). The VIC simulations driven by ***disturbed*** scenarios project a lower late fall and winter streamflow, with a delay in spring melt and subsequent increase in the pulse of peak streamflow during April–May. Recessional streamflow (May–Jul) is also slightly higher in the ***disturbed scenario*** than the ***climate-only*** scenario, resulting in greater water availability in summer (Fig. 3).

The mechanisms causing these differences in streamflow response to climate change and forest disturbance are illustrated in Figure 4 for forest dominant (more than 50%) regions. In the ***climate-only*** scenario, the streamflow response is dominated by the impact of temperature on snow pack. Warmer winter temperatures and reduced March snowfall (Fig. 4a) lead to a reduction in snow water equivalent, snow melt, and sublimation from both the pack and the canopy (Fig. 4b-e). Warmer temperatures,

earlier snow melt, and greater April rainfall subsequently lead to increases in soil evaporation (Fig. 4f) and transpiration (Fig. 4g) in the spring. Warmer temperatures also lead to increased canopy evaporation in late summer (Fig. 4h). However, the higher rates of soil evaporation and transpiration in the spring deplete the middle layer soil moisture (Fig. 4i), which diminishes their rates in the summer (Fig. 4f, g). In the fall, higher rainfall and warmer temperatures lead to greater soil evaporation but only

minimal replenishment of soil moisture. Both bottom layer soil moisture (Fig. 4j) and total (surface and subsurface) runoff (Fig. 4k) exhibit earlier and smaller peaks in the spring, reflecting the earlier melting of the reduced snow pack, and lower levels in summer due to greater ET in the spring.

Two factors in the ***disturbed*** scenario further impact the snow pack, partially compensating for the effects of climate change. First, replacement of forest with shrubs leads to an increase (relative to the ***climate-only*** scenario) in the on-the-ground snow pack accumulation (Fig. 4a), a prominent feature observed in disturbed forests across North America (Boon, 2007; Zou et al., 2010; Biederman et al.,

2015; Brown et al., 2014; Harpold et al., 2014). Shrubs in our VIC simulations have no canopy thus they have no mechanism to intercept snow. This disturbance-driven increase in on-the-ground snow pack partially compensates for the climate-induced streamflow increase during this time of the year. In addition, the larger on-the-ground snow pack and smaller canopy snow pack are accompanied by proportionally higher and lower rates of sublimation from the ground and canopy snow packs,

respectively, relative to ***climate-only*** (Fig. 4c, d). Second, the increase in snow on the ground caused by the higher shrub coverage leads to higher rates of snow melt (approximately 30%, Fig 4c), releasing water during the late spring and early summer (Apr–June, Fig 4k). The larger snow melt flux leads to substantial increases in transpiration in spring and early summer, not only relative to the ***climate-only*** scenario, but also relative to historical conditions (Fig. 4f). A similar increase in soil evaporation (Fig.

4f) does not occur due to the upper and middle soil layers already being saturated in the spring in the ***climate-only*** case (Fig. 4h). The high shrub transpiration rate in the ***disturbed*** scenario delays the larger snow melt flux in reaching the bottom soil layer (Fig. 4i), leading to a delayed peak in runoff (Fig. 4j, k). Meanwhile, LAI values are similar between shrubs and forests (Fig. 4l), indicating that the water and energy balance differences are due mainly to snow processes shifts between those two land cover types.

Differences in streamflow among ***disturbed*** and ***climate-only*** scenarios results are most notable at fine scales (100 km$^2$ to 3,000 km$^2$, 72 grid cells in the San Juan) where the forest cover dominates (more than 50%) the land cover (Fig. 5). As basin size increases (smaller circles in Fig. 5), and forest cover becomes sparser with respect to other types of land cover (e.g. mixed forest, deciduous, shrubs and grass covers), the differences between the ***climate-only*** scenarios and the ***disturbed*** scenarios begin to

decrease, which is corroborated by several other studies (Zhang et al., 2014; Anderegg et al., 2015). The difference between the two scenarios for the densest forest and smallest portion of the basin is approximately 20%, as observed for the larger circles in Fig. 5. This dynamic occurs even under changing precipitation and temperature projections that would otherwise cause increasing or decreasing

Deleted:

Moved (insertion) [2]

streamflow (observed in Fig. 5 colour ramps that change from red to blue for precipitation and different symbols for different temperature ranges). Even as the ESMs project increasing precipitation and temperature, we see the variability in responses through the cascade of scale and land cover variability. This finding is consistent with other studies that observed that variability (forest cover composition and topography) in the area and size of forest-shrub conversion can buffer responses of streamflow or ET shifts from climate change (Winkler et al., 2014; Harpold et al., 2014; Caldwell et al., 2016).

The dynamic between temperature, precipitation, and vegetation changes is examined in more detail by using a simple single cell analysis where climate and vegetation is altered in a sensitivity framework. Fig. 6 illustrates the single cell changes in terms of runoff from the grid cell to investigate the aridity effect. We can see that forests and shrubs act similarly in hot/warm and dry environments while the differences are more pronounced between forests and shrubs in wetter environments (Fig. 6a). For the vegetation changes, we see that changing forest structure (e.g., fraction of the canopy that may occur as a result of pest outbreaks) results in small shifts for forests but very large shifts for shrubs (Fig. 6b). For example, shrubs with 50% canopy spacing produce more water than the forests for the majority of runoff conditions (below the 3[rd] quantile, Fig 6b). On the other hand, under high runoff conditions (above the 3[rd] quantile, Fig 6b) shrubs act similarly to regularly spaced shrubs. These changes are on the same order as changes within the climate, generally speaking. Therefore, we hypothesize that larger changes in hydrology under disturbances are more likely to occur not from the forest disturbance itself, but the secondary effects such as regrowth of shrubs, the type of regrowth and the pattern of that regrowth on the landscape. This is why timescale is such an important consideration in studies of this nature, as regrowth patterns are key to understanding how water is partitioned across a disturbed landscape.

## 5    Discussion

Climate change in the CRB is anticipated to cause large impacts to water resources sustainability (Christensen and Lettenmaier, 2007; Rasmussen et al., 2014; Dawadi and Ahmad, 2012). However, to our knowledge few modelling studies have considered the impacts of climate change coupled with changes in vegetation (Buma and Livneh, 2015; Carroll et al., 2017; Pribulick et al., 2016). In this

study, we incorporated changes from the CMIP5 dynamic vegetation models and from an estimate of forest mortality (McDowell et al. 2016) to consider the impacts of both climate and vegetation changes on the water balance in a headwater system to the CRB, the San Juan River basin. We found that failing to consider climate change coupled with vegetation disturbances could result in a ~10% over-estimation

of the annual water availability for this basin. For a river system such as the Colorado that is already gravely stressed, 10% less water in the system may lead to significant water management challenges. And, considering seasonality in flows, the changes we illustrated in our monthly and seasonal scenarios indicate that less water will runoff during spring with more water arriving at peak melt. This could lead to water shortages and flooding and lead to planning issues for short term water delivery.

In this work, we considered not only the impacts of changes to streamflow but also the reasons why streamflow is changing. As in other studies, we found that at the size of the basin and the land cover variability can obscure the signal of change (Biederman et al., 2015; Penn et al., 2016). When we consider forested regions only, we are able to understand how and why streamflow is projected to change under the disturbed conditions. The main mechanism that is shifting streamflow is the manner in

which shrubs impact the water balance during the cold and warm seasons. Snow pack is retained further into the melt season and when snow starts to melt, it melts more quickly and results in higher peak flows. These peak flows, however, occur during a time that the shrubs are using the water, resulting in large transpiration losses. We found differences in canopy evaporation, soil evaporation, and sublimation, but these differences are quite small overall when comparing the overall volume of water

held back in the snowpack (SWE) and leaving via shrub transpiration. Overall, the differences in the mechanisms and delivery of the water changed, and this results in a 20% reduction in the amount of water that is available for runoff through the year, compared to the historical streamflow conditions for the forested regions of the San Juan River basin.

We also found that climate and disturbance have opposing influences on snow pack. In the early

season, the forest disturbance partially compensates for streamflow impacts caused by warmer temperatures (Fig. 3). We see that the shift towards an earlier snowmelt is compensated in part by the fact that shrubs held on to snow for longer into the snow melt season, releasing snow at a later date.

**Deleted:** (Pribulick et al., 2016)

This effect results in overall less water in the system and peak flows that occur on the same point in the year, but are higher in magnitude which results in lower soil moisture, such that the effect of having a late melting pack may not be beneficial for water resources. However, the snow-on-the-ground could have important consequences for early season energy balances.

The findings presented herein represent a deviation from the more broadly accepted viewpoint that forest disturbances will lead to reduced evapotranspiration and increased streamflow. Previous findings tend to be based on studies carried out over short time periods (first 1–2 year responses, < 5 year studies), pair-basin analyses in watersheds disturbed by clear cuts, and where climate variability may have obscured results (Brown et al., 2005). Studies pointing towards increased streamflow also broadly

found evapotranspiration from the canopy decreased, leading to an increase in runoff. However, work based on observations across scales and encompassing multiple disturbances indicates that the regrowth potential for understory, such as shrubs used in this study, is high and that the regrowth is a major control on water availability and direction of change for evapotranspiration and runoff (Caldwell et al., 2016; Biederman et al., 2014; Brown et al., 2014; Biederman et al., 2015; Pribulick et al., 2016).

Moreover, ecologists project that global forest covers are expected to decline and be replaced with species and understory compositions that are more water intensive. It is paramount therefore to treat regrowth correctly within models to align these two currently disparate principles and account accurately for changing streamflow. This is a fundamental issue because ESMs rely upon the research principles developed at plot-scale and watershed-scale observational studies and modelling work.

We also investigated the aridity effect upon water availability under forest disturbances. By comparing vegetation changes to climate shifts in a single cell, we highlighted the impacts of changing temperature and precipitation versus the impacts of changing forest cover properties. We found that the greatest differences in our results for forests and shrubs under climate change occurred under wet conditions, and vice versa for dry conditions. This may be due to the large component of water that is leaving via

evaporation in arid environments. As well, we find an important control on water availability in the shrub environments was the canopy spacing between the shrubs that led to changes in water balances and water partitioning within the environment. Thus, we believe that not only climate (i.e. aridity) but

**Deleted:**

**Deleted:** .

also vegetation characteristics (i.e. canopy spacing) may play a fundamental role in the impact of disturbances on water availability.

Our study did not incorporate fine-scale processes such as lateral flow, as these processes are not possible using the VIC modelling approach. Suggestions from studies examining vegetation structure

and patchiness illustrates that interconnected hillslope and riparian vegetation, such as shrublands, can receive water from wet upland catchment areas (Thompson et al., 2011). A recent study using the Parflow-CLM model, which incorporates lateral flow processes, found results that are very similar to our study findings (Pribulick et al., 2016). For example, Pribulick et al. (2016) show non-linear declines in streamflow in response to vegetation and temperature changes, with enhanced responses in

snowmelt-driven transects and where percent vegetation change was largest. Other research suggests that water availability declines, that plants adapt and becomes more efficient at using water (Troch et al., 2009), which is not incorporated in the approaches undertaken in this study and Pribulick et al. (2016). Additionally, VIC model parameters that impact changes in water balances have inherent uncertainty under climate change (Bennett et al., In review, 2017), and some of these parameters, such

as canopy overstory, are represented in a binary fashion, which is not necessarily indicative of the forest and shrublands mixtures observed during forest recovery. This is a clear example of improvements that could be made to the modeling approach that should be investigated in future work.

Our study also points to the challenges associated with incorporating spatially variable estimates of changing vegetation patterns. Although CMIP5 contains information from dynamic vegetation models,

we find a large disparity between the results from CMIP5 (**dynamic**) and the current estimates and cutting-edge research on forest cover mortality (**disturbed**); these methods are a scenario-style approach that does not necessary reflect future changing land cover conditions. The importance of accurately representing the effects that are observed at catchment scale and in offline models, such as VIC, cannot be understated. ESMs require the capability to incorporate these changes in a meaningful way that can

be validated using our current understanding of changes to forests in the Southwestern USA, and globally. This will enable the science community to accurately capture the range of responses and the impacts of changing climate and changing land cover on water resources. ESMs such as the Department

of Energy's ACME model have started down this path with the incorporation and testing of the FATES dynamic vegetation model (Fisher R. et al., 2015).

The differences in future streamflow projections in comparison to historical conditions we observed in this study are notable, and suggest that streamflow will decline by a minimum of ~10% for headwater systems of the Colorado. However, in the San Juan River basin forest cover accounts for only 17% of the area, and only 4% of these cells have greater than 80% forest cover. Impacts may be much larger for adjacent systems, such as the Gunnison. Future work will investigate the impacts of changing forest cover in the entire Colorado system, using sensitivity analysis to understand the cumulative effect of changing climate and vegetation cover in the Colorado.

## 6    Conclusion

The implications of declining streamflow due to forest disturbances in the San Juan River basin are significant for the CRB. Given that the San Juan basin is one of the least forested portions of the CRB and assuming similar rates of forest decline will occur nearby (Van Mantgem et al., 2009), other regions of the CRB are likely to experience streamflow declines of much higher severity due to changes in forest cover. Failure to adequately understand the direction and magnitude of changes as we do herein could have catastrophic consequences for those who rely on this resource. The basin—roughly 11% area of the continental United States—directly supports water supply for more than 30 million people, accounts for approximately 15% of the US's crops and livestock (Cohen et al., 2013), and 53 GW of power generation capacity (Buono and Eckstein, 2014; Cohen et al., 2013). Pressing work includes improvements in spatial representation of changing forest covers across the CRB, further investigations into drivers of change including an understanding of the role of aridity and variability, a stronger link between ecological dynamics and hydrological knowledge that is translated into models, and upward propagation of these findings into global and earth system models.

## 7    Acknowledgements

We acknowledge the support of John Abatzaglou and Katherine Hedgwick for providing us additional models not in the original MACA data set. We acknowledge the World Climate Research Programme's

Working Group on Coupled Modelling, which is responsible for CMIP, and we thank the climate modelling groups for producing and making available their model output. For CMIP, the U.S. Department of Energy's Program for Climate Model Diagnosis and Intercomparison provides coordinating support and led development of software infrastructure in partnership with the Global

5   Organization for Earth System Science Portals. Finally, Bennett, Solander, Middleton, McDowell and Xu acknowledge the Los Alamos National Lab's LDRD program for supporting this work. McDowell further acknowledges support of Pacific Northwest National Laboratories LDRD program. Bohn was supported by Grant 1216037 from the U.S. National Science Foundation (NSF) Science, Engineering and Education for Sustainability (SEES) Post-Doctoral Fellowship program and NSF Grant 1462086,

10  DMUU: Decision Centre for a Desert City III: Transformational Solutions for Urban Water Sustainability Transitions in the Colorado River Basin.

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

| | Objective Function | San Juan at Archuleta | San Juan at Bluff, UT |
|---|---|---|---|
| Calibration (2006-2010) | Volume Bias | 0.91 | 0.90 |
| | Nash Sutclife | 0.78 | 0.76 |
| | Nash Sutclife log | 0.77 | 0.75 |
| Validation (2001-2005) | Volume Bias | 0.93 | 0.93 |
| | Nash Sutclife | 0.83 | 0.60 |
| | Nash Sutclife log | 0.77 | 0.43 |
| Entire Period (2001-2010) | Volume Bias | 0.90 | 0.91 |
| | Nash Sutclife | 0.81 | 0.67 |
| | Nash Sutclife log | 0.78 | 0.58 |

**Table 2. IGBP classification names and codes from MODIS, our new classification, and remapped values (RM). Values for each ESM follows, MIROC-ESM, MPI-ESM-LR, HadGEM2-ES, and IPSL-CM5A-LR. For each ESM, the original code and the remapped (RM) values are given.**

| IGBP Classification | Code | Our Class | RM | MIROC-ESM | Code | RM | MPI-ESM-LR | Code | RM | HadGEM2-ES | Code | RM | IPSL-CM5B-LR | Code | RM |
|---|---|---|---|---|---|---|---|---|---|---|---|---|---|---|---|
| Evergreen Needleleaf forest | 1 | Evergreen forests | 1 | Tropical Forests | 1 | 1 | glacier | 1 | nc | broadleaf trees | 1 | 2 | Bare soil | 1 | 6 |
| Evergreen Broadleaf forest | 2 | Evergreen forests | 1 | Temperate/boreal evergreen | 2/4 | 1 | Tropical broadleaf evergreen | 2 | 1 | needleleaf trees | 2 | 1 | Tropical broad-leaved evergreen | 2 | 1 |
| Deciduous Needleleaf forest | 3 | Deciduous Forests | 2 | Temperate/boreal deciduous | 3/5 | 2 | Tropical broadleaf deciduous | 3 | 2 | C3/C4 grass | 3/4 | 3 | Tropical broad-leaved raingreen | 3 | 2 |
| Deciduous Broadleaf forest | 4 | Deciduous Forests | 2 | C3/C4 grass | 6/7 | 3 | Extra-tropical evergreen | 4 | 1 | shrubs | 5 | 4 | Temperate/boreal needleleaf evergreen | 4/7 | 1 |
| Mixed forest | 5 | Evergreen forests | 1 | Crop | 8 | 5 | extra-tropical deciduous | 5 | 2 | Urban/inland water/bare/ice | 6-9 | 6 | Temperate broad-leaved evergreen | 5 | 1 |
| Closed Shrublands | 6 | Shrubs | 4 | Pasture | 9 | 3 | raingreen shrubs | 6 | 4 | | | | Boreal needleleaf summergreen | 9 | 2 |
| Open Shrublands | 7 | Shrubs | 4 | Bare ground/Residual | 10/11 | 6 | decidous shrubs | 6 | 4 | | | | Temperate/boreal broad-leaved summergreen | 6/8 | 2 |
| Woody Savannas (40-60% tree cover) | 8 | Grass | 3 | | | | C3/C4 grass | 8/9 | 3 | | | | C3/C4 grass | 10/11 | 3 |
| Savannas (10-40% tree cover) | 9 | Grass | 3 | | | | C3/C4 pasture | 10/11 | 3 | | | | C3/C4 agriculture | 12/13 | 5 |
| Grasslands | 10 | Grass | 3 | | | | C3/C4 crops | 12/13 | 5 | | | | | | |
| Permanent wetlands (assuming like grass) | 11 | Grass | 3 | | | | | | | | | | | | |
| Croplands (corn) | 12 | Crops | 5 | | | | | | | | | | | | |
| Urban and built-up (assuming sparse grass) | 13 | Urban/ Water/ Bare | 6 | | | | | | | | | | | | |
| Cropland/Natural vegetation mosaic | 14 | Crops | 5 | | | | | | | | | | | | |
| Snow and ice (assuming sparse grass) | 15 | Urban/ Water/ Bare | 6 | | | | | | | | | | | | |
| Barren or sparsely vegetated (assuming sparse grass) | 16 | Urban/ Water/ Bare | 6 | | | | | | | | | | | | |

**Table 3.** *Average values for cells with forest cover greater than 50%. climate-only forest and shrub fractional values (top left), dynamic forest percentages for each climate model and all decades, and disturbed forests and shrubs for each decade. Below, Leaf Area Index (LAI, unitless), albedo (unitless) and canopy spacing (fraction) for all months.*

| climate-only forest climate-only shrubs | 0.70 (0.08) | Decade 10 | Decade 20 | Decade 30 | Decade 40 | Decade 50 | Decade 60 | Decade 70 | Decade 80 | Decade 90 | Decade 100 |
|---|---|---|---|---|---|---|---|---|---|---|---|
| *dynamic* forest (HAD) | | 0.69 | 0.69 | 0.68 | 0.67 | 0.65 | 0.63 | 0.61 | 0.59 | 0.57 | 0.56 |
| *dynamic* forest (MIR) | | 0.68 | 0.70 | 0.76 | 0.80 | 0.82 | 0.80 | 0.86 | 0.88 | 0.89 | 0.92 |
| *dynamic* forest (MPI) | | 0.71 | 0.70 | 0.70 | 0.69 | 0.70 | 0.71 | 0.68 | 0.67 | 0.66 | 0.66 |
| *dynamic* forest (IPSL) | | 0.70 | 0.70 | 0.70 | 0.70 | 0.70 | 0.71 | 0.71 | 0.71 | 0.71 | 0.71 |
| *dynamic* forest (HAD) | | 0.09 | 0.10 | 0.10 | 0.12 | 0.14 | 0.16 | 0.18 | 0.19 | 0.21 | 0.23 |
| *dynamic* forest (MIR) | | 0.11 | 0.09 | 0.07 | 0.05 | 0.05 | 0.06 | 0.04 | 0.03 | 0.03 | 0.02 |
| *dynamic* forest (MPI) | | 0.08 | 0.08 | 0.08 | 0.09 | 0.08 | 0.08 | 0.10 | 0.11 | 0.13 | 0.12 |
| *dynamic* forest (IPSL) | | 0.08 | 0.08 | 0.08 | 0.08 | 0.08 | 0.08 | 0.08 | 0.08 | 0.08 | 0.08 |
| disturbed forest | | 0.67 | 0.64 | 0.47 | 0.39 | 0.26 | 0.19 | 0.14 | 0.10 | 0.07 | 0.05 |
| disturbed shrubs | | 0.11 | 0.15 | 0.32 | 0.39 | 0.52 | 0.60 | 0.64 | 0.68 | 0.71 | 0.73 |

HAD=HadGEM2-ES, MIR=MIROC-ESM, MPI=MPI-ESM-LR, IPSL=IPSL-CM5-LR

| Months | Jan | Feb | Mar | Apr | May | Jun | Jul | Aug | Sep | Oct | Nov | Dec |
|---|---|---|---|---|---|---|---|---|---|---|---|---|
| LAI (unitless) | 0.55 (0.43) | 0.54 (0.42) | 0.57 (0.45) | 0.76 (0.66) | 1.20 (1.09) | 2.15 (2.01) | 2.17 (2.06) | 2.07 (1.98) | 1.92 (1.78) | 1.12 (1.04) | 0.75 (0.67) | 0.59 (0.48) |
| Albedo (unitless) | 0.21 (0.28) | 0.22 (0.28) | 0.20 (0.24) | 0.15 (0.17) | 0.11 (0.12) | 0.10 (0.11) | 0.11 (0.12) | 0.11 (0.11) | 0.10 (0.11) | 0.10 (0.11) | 0.12 (0.14) | 0.12 (0.14) |
| Canopy Spacing (fraction) | 0.10 (0.05) | 0.08 (0.04) | 0.09 (0.05) | 0.17 (0.13) | 0.39 (0.33) | 0.68 (0.61) | 0.75 (0.69) | 0.76 (0.70) | 0.69 (0.63) | 0.47 (0.41) | 0.33 (0.25) | 0.15 (0.09) |

1 **9 Figures**

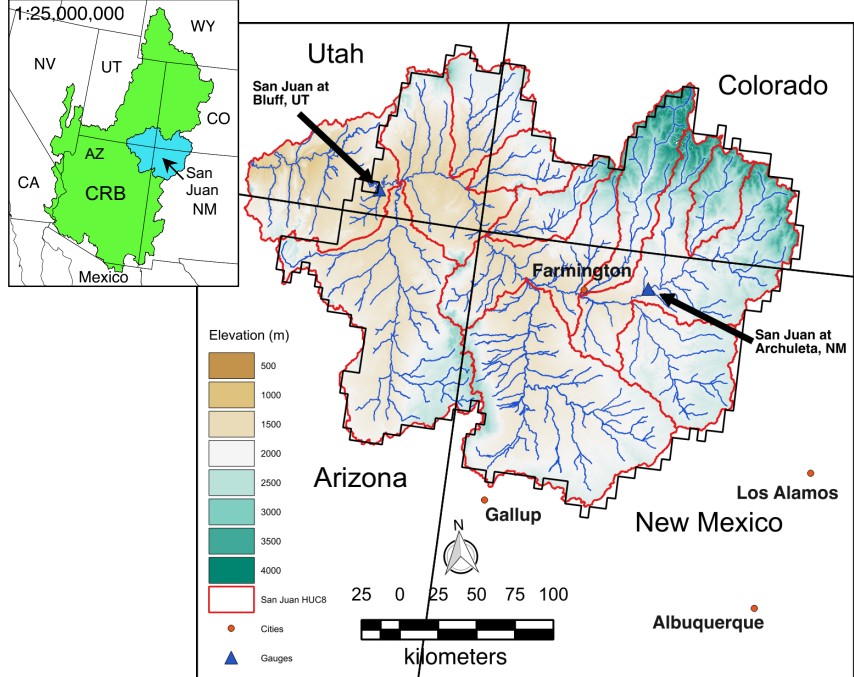

Figure 1. San Juan River basin (61,560 km$^2$) in the Colorado River watershed (634,150 km$^2$, inset), one of the most
important water resources in the western US. The San Juan River basin (outline of HUC8 watersheds shown in red)
spans the Four Corners region (CO, UT, AZ and NM) of the US, and supports multiple energy and water projects.
Naturalized flow gage sites (USBR) at San Juan at Bluff, UT and San Juan at Archuleta, NM are shown with closed blue
triangles.

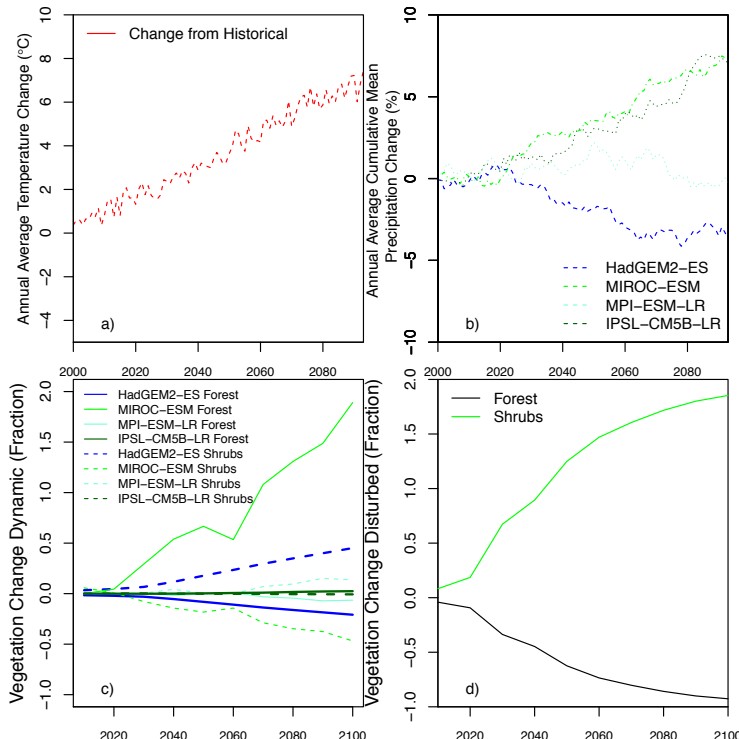

**Figure 2.** Temperature (°C, a), precipitation (%, b) and land cover (c and d) changes to the 2080s (2070-2099) averaged over the ESMs and spatially over the San Juan River basin (61,560 km$^2$ or 1570 grid cells). CMIP5 *dynamic* forest disturbances **for the four ESMs as a difference from the historical climatology (1970-1999)** illustrate more moderate change with HadGEM2-ES projecting the largest decline and MIROC-ESM projecting an increase in forest cover **(c)**. **The *disturbed* vegetation scenario based on McDowell et al. 2016 (d) illustrate strongly declining forest covers (>50% forest loss) and increased shrub covers.**

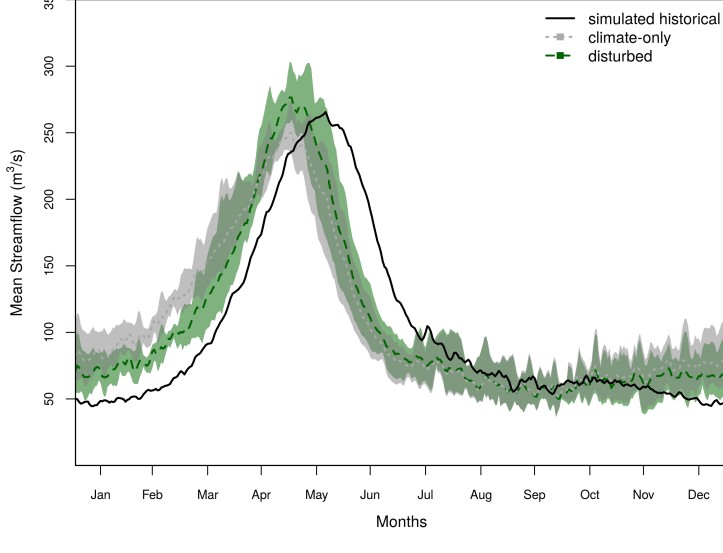

Figure 3. Monthly average future streamflows ($m^3$/s, 2070-2099) compared to historical (1970-1999) for the San Juan
River basin at Bluff, UT. The range of responses for each GCM is represented by the semi-transparent envelope around
the lines. The simulated historical streamflow is shown in black (single line, no envelope), the *climate-only* scenarios are
shown in grey (solid dark grey line, grey envelope), and the *disturbed* scenarios are shown in green (dashed dark green
line, green envelope).

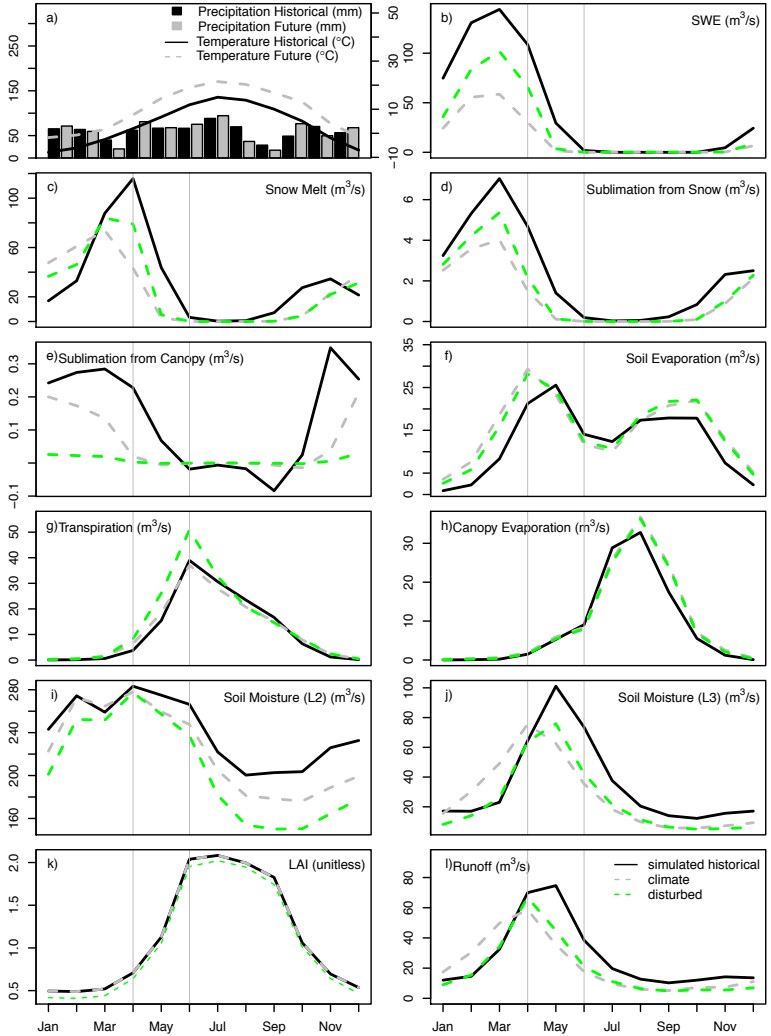

**Figure 4. Monthly water balance panel plot for forested (greater than 50%) regions in the San Juan river basin. Panels include a) precipitation (mm) and temperature (°C) for historical (1970-1999, black) and future (2070-2099, grey), snow water equivalent (SWE, m³/s, b), snow melt (m³/s, c), sublimation from snow (m³/s, d), sublimation from canopy (m³/s, e), soil evaporation (m³/s, f), transpiration (m³/s, g), canopy evaporation (m³/s, h), soil moisture from second soil later (L2, m³/s, i), soil moisture from the third soil layer (L3, m³/s, j), leaf area index (LAI, unitless, k), and runoff (m³/s, l), for simulated historical (black line), climate-only (grey line) and disturbed scenarios (green line). The months of April and June are indicated with vertical grey lines.**

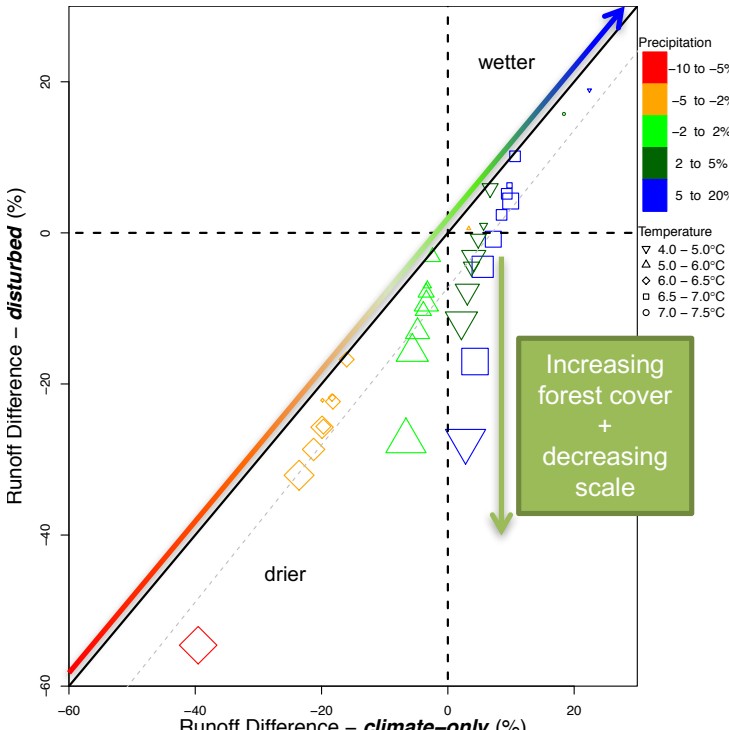

**Figure 5. Runoff differences for the** *climate-only* **scenario (x-axis) plotted against** *disturbed* **scenario (y-axis).** **Increasing**
**sizes indicate results from the entire San Juan River basin (smallest), to all forests (next smallest size), and then forests**
**with greater than 10%, 50%, 70% and 90% coverage (largest).** **All changes are shown as projected differences from the**
**historical (in % change).**

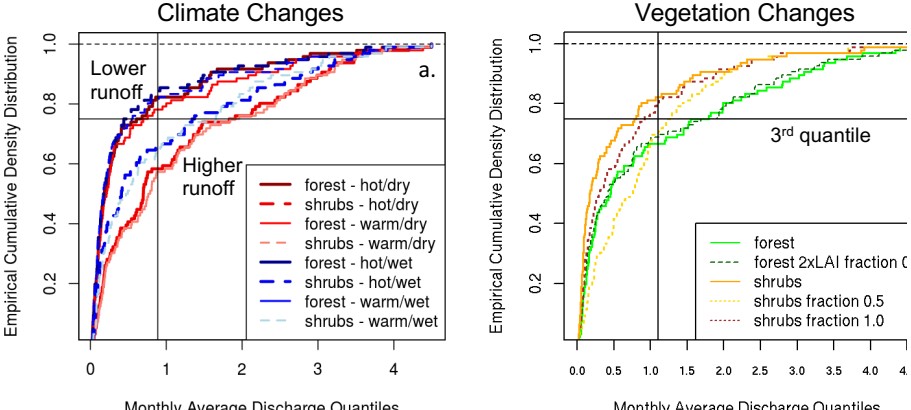

Figure 6. Cumulative density distributions are used here to illustrate the variations in (a) climate and (b) vegetation parameterizations for a single cell in the San Juan River basin. Climate alterations are shown for a hot-wet, hot-dry, warm-wet and warm-dry scenarios for both forest and shrubs. For the vegetation parameterization, forests, forests with two times leaf area index (LAI) and canopy fraction of 0.5, shrubs, and shrubs with fraction of 0.5 and 1.0. A shrub fraction of 1 indicates that there are no spaces between the plants. The third quantile in the average of the distributions is indicated in each plot with intersecting lines.

