# Peer review of "Katrina E. Bennett1, Theodore Bohn2,3, Kurt Solander1, Nathan G. McDowell1,5, Chonggang Xu1, Enrique Vivoni3,4 and Richard S. Middleton1"

_Hydrology and Earth System Sciences, 2017_

## Referee Comment (RC1) · Anonymous Referee #1 · 3 Sep 2017

"Climate change and climate-driven disturbances in the San Juan River sub-basin of the Colorado River" submitted by Bennet et al. addresses the timely and important question of the interrelated influences of vegetation and climate change on a Colorado River headwater's system. I found the title to be appropriate and the abstract to represent the discussion presented in the manuscript. While I generally agree that vegetation dynamic may present an important complication to modeling future climate states, I feel the authors did not clearly explain the mechanisms driving the modeled change or thoroughly fit their work into a greater body of growing literature as summarized in my general comments below.

[Figure]

1. The vegetation properties and dynamics are not clear, particularly for a reader that is not familiar with the dynamic vegetation processes in Earth Systems Models (ESMs). The authors should provide additional details on how vegetation dynamics are modeled in the ESMs used here, as well as in the VIC simulations of vegetation change. This description should include the range of relevant vegetation parameters for each scenario/land cover classificiation (LAI, coverage, etc.) Ultimately, this discussion should also support a better description of the mechanisms behind the modeled hydrologic change. For example, the authors state that LAI values are similar between shrubs and forests (pg 10 ln 16), and that the changes in water and energy balances are therefore related to changes in snow processes. LAI is known to have a strong control on snow processes, so in the absence of LAI differences, the authors should explain what physical vegetation characteristics are driving these changes.

2. The authors failed to cite a substantial number of recent references on vegetation and climate change in the Rocky Mountains, often relying on references from other regions such as Canada and Alaska. The differences in aridity and evaporative demand suggest regional references are more appropriate. (For example see Pribulek et al 2016 and Carroll et al. 2017 for additional modeling studies on vegetation and climate change effects on hydrology using more integrated modeling approaches, Penn et al 2016 for a modeling study of the effect of vegetation change across scales, Livneh et al. 2015 for another bark beetle modeling study that shows muted streamflow effects with regrowth, and Bearup et al. 2016 for a paper on vegetation effects on changes in streamflow partitioning). These references may also help to support a discussion on the importance of groundwater and evapotranspiration in this system and across scales.

Technical Corrections: Pg 4 Ln 16-18: Check section numbers Pg 7 Ln 10-13: At what timescales are the model results and observations compared for calculation of NSE? Hourly? Figure 3: It is not clear what the light gray shading is or why there is a gap in the dark grey shading near peak streamflow (i.e. late April). Figure 3 Caption: Clarify

if historical period is from model runs or observations (throughout). Figure 4: It would be interesting to see how rain and snow is partitioned differently due to temperature change in these scenarios, either here or in another figure. Also, the axes units are not provided.

———————————————

---

## Referee Comment (RC2) · Anonymous Referee #1 · 3 Sep 2017

REFERENCES:

Bearup, L.A., Maxwell, R.M. and McCray, J.E., 2016. Hillslope response to insect-induced land-cover change: an integrated model of end-member mixing. Ecohydrology, 9(2), pp.195-203.

Bearup, L.A., Maxwell, R.M., Clow, D.W. and McCray, J.E., 2014. Hydrological effects of forest transpiration loss in bark beetle-impacted watersheds. Nature Climate Change, 4(6), pp.481-486.

Carroll, R.W., Huntington, J.L., Snyder, K.A., Niswonger, R.G., Morton, C. and Stringham, T.K., 2017. Evaluating mountain meadow groundwater response to Pinyon-

Juniper and temperature in a great basin watershed. Ecohydrology, 10(1).

Livneh, B., Deems, J.S., Buma, B., Barsugli, J.J., Schneider, D., Molotch, N.P., Wolter, K. and Wessman, C.A., 2015. Catchment response to bark beetle outbreak and dust-on-snow in the Colorado Rocky Mountains. Journal of Hydrology, 523, pp.196-210.

Penn, C.A., Bearup, L.A., Maxwell, R.M. and Clow, D.W., 2016. Numerical experiments to explain multiscale hydrological responses to mountain pine beetle tree mortality in a headwater watershed. Water Resources Research, 52(4), pp.3143-3161.

Pribulick, C.E., Foster, L.M., Bearup, L.A., Navarre-Sitchler, A.K., Williams, K.H., Carroll, R.W. and Maxwell, R.M., 2016. Contrasting the hydrologic response due to land cover and climate change in a mountain headwaters system. Ecohydrology, 9(8), pp.1431-1438.

---

## Referee Comment (RC3) · Anonymous Referee #2 · 6 Oct 2017

The authors present an interesting analysis into the potential combined effects of climate and land-cover change in a major tributary of the Colorado River. While the paper has sufficient novelty to be of interest to the community, several key oversights need to be addressed. For this reason, I recommend the paper undergo minor revisions prior to publication.

Major points

1. The authors conclude that understory regrowth leads to reduced streamflows. While this is a logical conclusion, additional discussion into other important mechanistic changes is warranted. First, if the model used by the authors doesn't account

for lateral flow (i.e. is it a 1-D model?), then the authors need to acknowledge lack of process representation important for vegetation/hydrology interaction. For example, low-lying vegetation can receive water from wetter-headwaters areas of a catchment (e.g. Troch et al., 2009, Hydrologic Processes; Thompson et al., 2011, WRR). If these dynamics are ignored, then at minimum the authors need to acknowledge how the findings of these earlier works may impact the results in their manuscript.

2. An important missing piece is a justification for the settings used in the 'disturbed' forest scenario. While LAI was changed, the authors need to provide more justification for why the vegetation was modified the way that it was, and how this compares to what previous modeling studies have done. For example, other studies have explored changes in canopy transmissivity associated with forest disturbance (e.g. Bewley et al., 2010, J.Hydrology—the authors cite this paper, but do not reference the important 'calibration' to transmissivity that was done) involving calibration of forest parameters to observations, or remote sensing (Baker et al., 2017, RSE)while others have also modified the stomatal resistance, which is critical for accurately modulating ET, that is to say that modifying LAI alone may result in an inaccurate change to the total ET (e.g. Livneh et al.,; J.Hydrology).

3. Along the lines of the previous comment, it is unclear whether the authors validated their model beyond historical streamflow comparison. While streamflow comparison is important, a validation of the impact of imposed vegetation changes is warranted to ensure the settings and modifications are realistic, while challenging, this could be done on historical observations of key model structural components using in situ and remotely sensed observations. Alternatively, if validating the model settings is outside the scope of the present manuscript, then the authors need to clearly state this as a limitation of their study in the discussion.

4. Research now exists that suggests that forests disturbance can cause both increases and decreases in streamflow. While the authors cite part of the literature, they ignore important foundational publications on this topic (see references within Adams

et al., 2012, Ecohydrology). Newer research also exists that uses large-sample catchments to study the effects of forest disturbance and should be referenced here (Buma et al., 2017; ERL), so as to put the findings in a clearer context relative to the literature.

5. More details into the climate of the study basin is warranted, e.g. mean annual precipitation, temperature (include summary numbers in addition to figure 4), mean elevation of the basin, etc. Most importantly, the authors need to comment on how representative this semi-arid basin is of the Colorado basin as a whole and whether the results here are indicative of what other parts of the world might see, or whether this climate/landscape is sufficiently unique.

Minor

Figure 3: this is a nice figure, however the axis labels are way too small and need to be increased, as does the font size in the legend. Furthermore, the green shading should be included in the legend.

Figure 4, the lines are too thin and difficult to see. The authors should increase the line thickness by at least a factor of 2.

Figure 5 is very difficult to follow. First, it seems to me that including multiple colored circles directly on the plot is confusing, since it's unclear whether these are a part of the plot or a legend. To overcome this, the authors should add a color bar and remove these circles. Second, the circles in the upper left of the plot obfuscate the graph, since I cannot find any other circles (other than a small one in the upper right), so instead the authors should state in the caption that the size of the symbols corresponds to the percent forest, there could be a length bar in the legend to show this, but the circles are too distracting/confusing. Lastly, the temperature description in the top center of the plot belongs in a legend.

Figure 6: The lines are very difficult to distinguish, particularly in panel (a). I recommend that the authors avoid using dashed and solid lines of the same color in such

close proximity, i.e. use lines of different colors.

---

## Author Comment (AC1) · 31 Oct 2017

"Climate change and climate-driven disturbances in the San Juan River sub-basin of the Colorado River" submitted by Bennett et al. addresses the timely and important question of the interrelated influences of vegetation and climate change on a Colorado River headwater's system. I found the title to be appropriate and the abstract to represent the discussion presented in the manuscript. While I generally agree that vegetation dynamic may present an important complication to modeling future climate states, I feel the authors did not clearly explain the mechanisms driving the modeled change or thoroughly fit their work into a greater body of growing literature as summarized in my general comments below.

1. The vegetation properties and dynamics are not clear, particularly for a reader that is not familiar with the dynamic vegetation processes in Earth Systems Models (ESMs). The authors should provide additional details on how vegetation dynamics are modeled in the ESMs used here, as well as in the VIC simulations of vegetation change. This description should include the range of relevant vegetation parameters for each scenario/land cover classification (LAI, coverage, etc.) Ultimately, this discussion should also support a better description of the mechanisms behind the modeled hydrologic change. For example, the authors state that LAI values are similar between shrubs and forests (pg 10 ln 16), and that the changes in water and energy balances are therefore related to changes in snow processes. LAI is known to have a strong control on snow processes, so in the absence of LAI differences, the authors should explain what physical vegetation characteristics are driving these changes.

Response: This critique appears to apply both to ESMs in general, and to VIC as employed in this study. Regarding ESMs in general, we have now added an additional paragraph discussing the limitations of current ESMs and their dynamic vegetation processes on page 4. As we mention in the manuscript and now repeat in the newly added text, the ESM vegetation changes we include in this study are limited because the limitations within ESM projections of vegetation dynamics. This is why we include a second approach (McDowell et al., 2016) to estimate future changes to vegetation.

Regarding the range of relevant vegetation parameters for each scenario used in VIC, we now include, in addition to LAI being a panel in Figure 4, a description of the average forest and shrub fractions under each scenario and LAI, albedo and canopy fraction for average forest and shrub as Table 3 in the manuscript. The way in which we modified forests within the VIC model meant that we only changed the fraction of the forests in each grid cell, as described in the Methods section 3.3. LAI, albedo and canopy fraction values do not change through the scenarios, but as LAI, for example, is different cell-by-cell for shrubs versus forests, different values are applied when the model is run using different fractions of these land cover types.

Regarding LAI values being similar between forests and shrubs for the forested regions of the watershed (greater than 50% forest cover). The MOD15A2 (Myneni et al., 2002; Schaaf et al., 2002; Huete et al., 2002) data product, which is what our Leaf Area Index estimates are based on, provides shrub LAI values that are similar to forests. Note that this isn't an issue if we were to examine the entire San Juan River basin as a single unit, for instance. In that case, the LAI for shrubs is approximately half or less than that of the forests. As LAI isn't changing very much in our scenarios, the main differences between forest and shrublands is (a) overstory vs. no overstory, (b) deeper root depths, and (c) differences in sensitivy of stomatal resistance to solar radiation and sensitivity of canopy resistance to LAI. These are all vegetation library parameters. We think (a) is the most important of these, as overstory is what creates canopy snowpack, which has a big impact on how long it takes the snowpack to melt. We describe

these physical changes leading to streamflow shifts on page 11. However, this may be a model weakness that is worth noting in the paper and we have done so in the updated Discussion section of the manuscript.

2. The authors failed to cite a substantial number of recent references on vegetation and climate change in the Rocky Mountains, often relying on references from other regions such as Canada and Alaska. The differences in aridity and evaporative demand suggest regional references are more appropriate. (For example see Pribulek et al 2016 and Carroll et al. 2017 for additional modeling studies on vegetation and climate change effects on hydrology using more integrated modeling approaches, Penn et al. 2016 for a modeling study of the effect of vegetation change across scales, Livneh et al. 2015 for another bark beetle modeling study that shows muted streamflow effects with regrowth, and Bearup et al. 2016 for a paper on vegetation effects on changes in streamflow partitioning). These references may also help to support a discussion on the importance of groundwater and evapotranspiration in this system and across scales.

Response: We have added the following references to the manuscript as suggested by the reviewer. We have added text and cited Livneh et al. 2015 on page 3, lines 23-24. We have also added the recently released Buma et al. 2017 to this paragraph as per comments by Reviewer #2. Bearup et al. (2014 and 2016) were added to the same paragraph and in Section 2 of the paper. Penn et al. 2016 was added to the Introduction and also presented in Section 2 and the Discussion. Pribulek et al. 2016 and Carroll et al. 2017 were cited in the Discussion section. One important point to note is that this paper is focused on a study of landscape change and disturbances (drought, forest forests, pests) under climate change. As such, there are a number of fundamental differences in this study in comparison with previous work, the most important difference is time scale. In this work, we consider the time scale of multiple decades and thus we account for replacement of *forests with shrublands* occurring in these watersheds. As we note, this regrowth dominates streamflow response. Our other main point is that there are important scalar effects occurring when considering the watershed response to impacts by forest cover disturbances; these scalar effects are related to the amount of forest impacted in relation to other land cover types in the basin and the size of the basin. We have altered the manuscript to make these points more clearly and used the above citations to support the points. Please see revised version of the manuscript.

Technical Corrections:
Pg 4 Ln 16-18: Check section numbers

Response: Thank you, we have corrected the section numbers where they were incorrect.

Pg 7 Ln 10-13: At what timescales are the model results and observations compared for calculation of NSE? Hourly?

Response: We have added in "monthly" to the sentence to indicate the simulated-observed calibration/validation periods. Only monthly naturalized streamflow data is available for calibration and validation hence this is the time scale we calibrated for. The caption of Table 1 has also been edited to make it clear that monthly data was used.

Figure 3: It is not clear what the light gray shading is or why there is a gap in the dark grey shading near peak streamflow (i.e. late April).

Response: We thank the reviewer for this comment and the attention to detail. We have removed the light grey shading background to make it clearer and changed the legend in this

figure to indicate that the envelopes of color represent the range of results across ESMs. We have also updated the caption of this figure appropriately.

Figure 3 Caption: Clarify if historical period is from model runs or observations (throughout).
Response: Thank you. We have changed the legend to read "simulated historical".

Figure 4: It would be interesting to see how rain and snow is partitioned differently due to temperature change in these scenarios, either here or in another figure. Also, the axes units are not provided.

Response: This is an interesting comment. We did not output separate rain and snow simulation results and thus we would need to rerun all our simulations in order to answer this question. However, we do believe that mid-winter warming events or rain-to-snow transitions are occurring under future scenarios in the San Juan River basin and this is the focus of alternate work on climate change impacts we are pursuing. However, the focus of this study is on the vegetation scenarios versus climate change signals, and since all scenarios have the same four climate model inputs, all scenarios have the same partitioning of snow/rain. Therefore, this is not a contributing factor to the differences between the simulations we are exploring in this manuscript. Thus, we have not altered Figure 4 panel a to add in partitioning of precipitation between rain and snow. We have added axes units to the panel titles.

References:
Bearup, L.A., Maxwell, R.M. and McCray, J.E., 2016. Hillslope response to insect induced land-cover change: an integrated model of end-member mixing. Ecohydrology, 9(2), pp.195-203.

Bearup, L.A., Maxwell, R.M., Clow, D.W. and McCray, J.E., 2014. Hydrological effects of forest transpiration loss in bark beetle-impacted watersheds. Nature Climate Change, 4(6), pp.481-486.

Carroll, R.W., Huntington, J.L., Snyder, K.A., Niswonger, R.G., Morton, C. and Stringham, T.K., 2017. Evaluating mountain meadow groundwater response to Pinyon-Juniper and temperature in a great basin watershed. Ecohydrology, 10(1).

Livneh, B., Deems, J.S., Buma, B., Barsugli, J.J., Schneider, D., Molotch, N.P., Wolter, K. and Wessman, C.A., 2015. Catchment response to bark beetle outbreak and dust on-snow in the Colorado Rocky Mountains. Journal of Hydrology, 523, pp.196-210.

Penn, C.A., Bearup, L.A., Maxwell, R.M. and Clow, D.W., 2016. Numerical experiments to explain multiscale hydrological responses to mountain pine beetle tree mortality in a headwater watershed. Water Resources Research, 52(4), pp.3143-3161.

Pribulick, C.E., Foster, L.M., Bearup, L.A., Navarre-Sitchler, A.K., Williams, K.H., Carroll, R.W. and Maxwell, R.M., 2016. Contrasting the hydrologic response due to land cover and climate change in a mountain headwaters system. Ecohydrology, 9(8), pp.1431-1438.

**Citations:**

Huete, A., Didan, K., Miura, T., Rodriguez, E. P., Gao, X., and Ferreira, L. G.: Overview of the radiometric and biophysical performance of the MODIS vegetation indices, Remote sensing of environment, 83, 195-213, 2002.

McDowell, N. G., Williams, A. P., Xu, C., Pockman, W. T., Dickman, L. T., Sevanto, S., Pangle, R., Limousin, J., Plaut, J., Mackay, D. S., Ogee, J., Domec, J. C., Allen, C. D., Fisher, R. A., Jiang, X., Muss, J. D., Breshears, D. D., Rauscher, S. A., and Koven, C.: Multi-scale predictions of massive conifer mortality due to chronic temperature rise, Nature Clim. Change, 6, 295-300, 10.1038/nclimate2873, 2016.

Myneni, R. B., Hoffman, S., Knyazikhin, Y., Privette, J. L., Glassy, J., Tian, Y., Wang, Y., Song, X., Zhang, Y., Smith, G. R., Lotsch, A., Friedl, M., Morisette, J. T., Votava, P., Nemani, R. R., and Running, S. W.: Global products of vegetation leaf area and fraction absorbed PAR from year one of MODIS data, Remote Sensing of Environment, 83, 214-231, http://dx.doi.org/10.1016/S0034-4257(02)00074-3, 2002.

Schaaf, C. B., Gao, F., Strahler, A. H., Lucht, W., Li, X., Tsang, T., Strugnell, N. C., Zhang, X., Jin, Y., and Muller, J.-P.: First operational BRDF, albedo nadir reflectance products from MODIS, Remote sensing of Environment, 83, 135-148, 2002.

---

## Author Comment (AC2) · 31 Oct 2017

The authors present an interesting analysis into the potential combined effects of climate and land-cover change in a major tributary of the Colorado River. While the paper has sufficient novelty to be of interest to the community, several key oversights need to be addressed. For this reason, I recommend the paper undergo minor revisions prior to publication.

**Major points**

1. The authors conclude that understory regrowth leads to reduced streamflows. While this is a logical conclusion, additional discussion into other important mechanistic changes is warranted. First, if the model used by the authors doesn't account for lateral flow (i.e. is it a 1-D model?), then the authors need to acknowledge lack of process representation important for vegetation/hydrology interaction. For example, low-lying vegetation can receive water from wetter-headwaters areas of a catchment (e.g. Troch et al., 2009, Hydrologic Processes; Thompson et al., 2011, WRR). If these dynamics are ignored, then at minimum the authors need to acknowledge how the findings of these earlier works may impact the results in their manuscript.

Response: Yes, the model we used in this work, the Variable Infiltration Capacity (VIC) hydrologic model, does not include lateral flow. This is largely because VIC was developed as a land surface scheme for regional studies of climate change impacts and it is not meant for hillslope or plot-scale studies. In this work, we are considering regional scale responses of streamflow to climate change and land cover disturbances thus we feel it is an appropriate model for our study. This study is part of a larger project focused on understanding the kinds of questions regarding lateral flow that the reviewer raises, and we aim to investigate this question specifically in future efforts. We have added a section discussing the lack of lateral flow consideration in our study to the Discussion section and have cited the papers suggested by the reviewer, along with other citations from a recent study by Pribulick et al. 2016 (suggested by Reviewer #1) that does incorporate lateral flow. Please see track changed version of the manuscript.

2. An important missing piece is a justification for the settings used in the 'disturbed' forest scenario. While LAI was changed, the authors need to provide more justification for why the vegetation was modified the way that it was, and how this compares to what previous modeling studies have done. For example, other studies have explored changes in canopy transmissivity associated with forest disturbance (e.g. Bewley et al., 2010, J.Hydrology. The authors cite this paper, but do not reference the important 'calibration' to transmissivity that was done) involving calibration of forest parameters to observations, or remote sensing (Baker et al., 2017, RSE) while others have also modified the stomatal resistance, which is critical for accurately modulating ET, that is to say that modifying LAI alone may result in an inaccurate change to the total ET (e.g. Livneh et al.,; J.Hydrology).

Response: In our study, we did not modify LAI. We modified the forest cover percentages, and kept LAI, albedo and vegetation canopy spacing the same as these will be altered in VIC as the forest cover percentages change in our scenarios. One of the reasons why we believe shrublands respond the way they do, is that shrubland LAI is in fact, similar, to forest LAI in the forested, upland headwater catchment regions of the San Juan River basin. Please see our response to Reviewer #1, question #1/

3. Along the lines of the previous comment, it is unclear whether the authors validated their model beyond historical streamflow comparison. While streamflow comparison is important, a

validation of the impact of imposed vegetation changes is warranted to ensure the settings and modifications are realistic, while challenging, this could be done on historical observations of key model structural components using in situ and remotely sensed observations. Alternatively, if validating the model settings is outside the scope of the present manuscript, then the authors need to clearly state this as a

limitation of their study in the discussion.

Response: We based our study of vegetation changes on two sources of information, dynamic vegetation in CMIP5 and the changes in vegetation disturbances determined by McDowell et al. 2015. One of the challenges that we face is that historical comparisons of vegetation disturbances are not valid for future, no-analogue conditions. We do not know what these future changes are going to look like, hence we examine a scenario-style approach to changing forest to simulate disturbances and then discuss the responses. We do not attempt to say that our methods are correct, and we refer to them only as projections of the future changes. We have added a line into the Discussion on page 15 that clearly states this point.

4. Research now exists that suggests that forests disturbance can cause both increases and decreases in streamflow. While the authors cite part of the literature, they ignore important foundational publications on this topic (see references within Adams et al., 2012, Ecohydrology). Newer research also exists that uses large-sample catchments to study the effects of forest disturbance and should be referenced here (Buma et al., 2017; ERL), so as to put the findings in a clearer context relative to the literature.

Response: We do discuss this in our manuscript. We draw the reviewer's attention to page 3,. "To date, predictions of future streamflow in forested river basins under future changes in climate and land cover have exhibited wide disagreement as to the strength and even the sign of change." We cite in this paragraph Adams et al 2012, and Guardiola-Claramonte et al., 2011, for example. In Adams et al. 2012, Table 2 lists studies that included hydrological response to tree die-off. This table includes one study we did not reference that cites mostly decreasing streamflow (1 out of 8 catchments) and no change (7 out of 8 catchments), Somer et al. 2010. This is also discussed in McDowell et al. (In press, 2017), and we have added this citation to the paragraph. We have now added the reference to our paragraph, and specifically noted Adams et al.'s Table 2. We have modified the text in the Introduction and have also added the newly published Buma et al. 2017 to our citations in this paragraph.

5. More details into the climate of the study basin is warranted, e.g. mean annual precipitation, temperature (include summary numbers in addition to figure 4), mean elevation of the basin, etc. Most importantly, the authors need to comment on how representative this semi-arid basin is of the Colorado basin as a whole and whether the results here are indicative of what other parts of the world might see, or whether this climate/landscape is sufficiently unique.

Response: We have added mean annual precipitation and January/July temperature to the Study Site description. Figure 1 provides the elevation across the basin, so we have not added elevation statistics to this paragraph. The San Juan can be thought of as a microcosm of the Colorado, because it contains both high elevation mountains representative of the upper basin, and the flat lowland, semi-arid environment indicative of the lower basin. We have amended our Study Site description to read: "The San Juan basin captures the diversity present across the CRB. For instance, high elevation (> 4000 m) Colorado mountain ranges and large, snowmelt driven rivers comprise the upper San Juan basin. The lower San Juan basin, located in New Mexico and Arizona, is flat, semi-arid and representative of the lower Colorado, with intermittent

streams that drain into the main tributary of the San Juan during the summer when they are charged by summer monsoonal rains."

**Minor**

Figure 3: this is a nice figure, however the axis labels are way too small and need to be increased, as does the font size in the legend. Furthermore, the green shading should be included in the legend.

Response: Thank you. We have adjusted this figure to address your comments and Reviewer #1's comments. The figures axes labels are now larger and the legend font is larger. The green and grey shading are now included in the legend.

Figure 4, the lines are too thin and difficult to see. The authors should increase the line thickness by at least a factor of 2.

Response: Thank you. We have adjusted the line thickness and added axes units to this figure.

Figure 5 is very difficult to follow. First, it seems to me that including multiple colored circles directly on the plot is confusing, since it's unclear whether these are a part of the plot or a legend. To overcome this, the authors should add a color bar and remove these circles. Second, the circles in the upper left of the plot obfuscate the graph, since I cannot find any other circles (other than a small one in the upper right), so instead the authors should state in the caption that the size of the symbols corresponds to the percent forest, there could be a length bar in the legend to show this, but the circles are too distracting/confusing. Lastly, the temperature description in the top center of the plot belongs in a legend.

Response: Thank you. We have amended the figure and we think it is vastly improved.

Figure 6: The lines are very difficult to distinguish, particularly in panel (a). I recommend that the authors avoid using dashed and solid lines of the same color in such close proximity, i.e. use lines of different colors.

Response: Thank you. We have amended the figure panel a to add more colors to differentiate the lines.

**Citations**

McDowell, N. G., Michaletz, S., Bennett, K. E., Solander, K., Xu, C., Maxwell, R. M., Allen, C. D., and Middleton, R. S.: Predicting Chronic Climate-Driven Disturbances and Their Mitigation, Trends in Ecology & Evolution (TREE), In press, 2017.

---

## Author Response (AR2)

December 9th, 2017

Dear Reviewers and HESS Editors,

We have address all the comments, including all the technical updates to Figures, in this version of the manuscript.

Thank you very much.

Sincerely,

Katrina Bennett